# Householder-Diagonalized Linear Attention (HDLA): Utilizing Rank-Enhanced Decay Mechanism for Efficient Sequence Modeling

**Jiefu Zhang**[1,5]*, **Zhen Qin**[2]*, **Jiabo Tong**[1,3], **Shijie Mei**[1], **Jiakui Hu**[4], **Yuqi Pan**[1], **Anjie Hu**[1,3], **Man Yao**[1], **Bo XU**[1], **Guoqi Li**[1]†

[1] Institute of Automation, Chinese Academy of Sciences
[2] Taptap
[3] Beijing Zhongguancun Academy
[4] Peking University
[5] School of Advanced Interdisciplinary Sciences, University of Chinese Academy of Sciences

## Abstract

Linear attention mechanisms have emerged as efficient alternatives to Softmax attention, exhibiting steady improvements in language modeling capabilities driven by increasingly sophisticated designs for decay matrices—though their structural complexity has typically been limited to the Diagonal-Plus-Rank-1 level. To further advance the understanding and capabilities of linear attention via more complex decay structures, this work makes two primary contributions: (1) We propose the HDLA linear attention mechanism, which utilizes efficient matrix decomposition to achieve a Diagonal-Plus-Rank-2 structure, thereby extending the decay matrix to a broader, more expressive, rank-enhanced and structured class. (2) We propose a more general chunk-wise parallel algorithm that accommodates both diagonal-plus-rank-$r_{ab}$ decay structure and key-value outer products of rank $r_{kv}$, thus providing a versatile foundation for future research. Comprehensive experiments demonstrate that, compared to linear attention baselines, HDLA sets new SOTA results on language modeling and retrieval tasks at 2.8B parameter scale, delivers at most 80% and 58.2% performance gains over baselines on retrieval-based MQAR and RULER tasks, and achieves an average score improvement of 4.39–7.66 on the synthetic MAD benchmark, respectively. Our proposed HDLA model, as well as the rank-generalized chunk-wise parallel algorithm, together provide a versatile algorithmic foundation and promising research prospects for the design of rank-enhanced, structured linear attention mechanisms.

## 1 Introduction

Softmax attention, the core component of the Transformer (Vaswani et al., 2017), exhibits superior token mixing capabilities (Tolstikhin et al., 2021; Yu et al., 2022) and supports highly efficient parallel training (Dao et al., 2022). However, it is severely limited in long context scenarios, by quadratic time complexity and a key-value (KV) cache that grows linearly with the sequence length.

Linear attention presents an efficient alternative to softmax attention by reducing the time complexity to $O(n)$ and compressing the infinite key-value sequences into a fixed-size hidden state (Katharopoulos et al., 2020). Not only does it demonstrate great research potential, but the hybrid architecture combining linear and softmax attention in 7:1 ratio has been successfully deployed as the foundational framework for large language models (LLMs) in practical applications, achieving exceptional throughput and advanced long-context reasoning capabilities (MiniMax et al., 2025). Through progressively more sophisticated hidden state decay mechanisms, linear attention has steadily improved

---

*Equal contribution for Jiefu Zhang and Zhen Qin. This work was completed while Zhen Qin was a researcher at Taptap. Code is available at `https://github.com/Zhangjiefu777/HDLA-Impl`.

†Corresponding author, guoqi.li@ia.ac.cn

its language modeling performance. Nevertheless, a series of recent works—including DeltaNet (Yang et al., 2024b), Gated DeltaNet (Yang et al., 2025), TTT-Linear (Sun et al., 2024)—restrict the structural complexity of their decay matrices to at most Diagonal-Plus-Rank-1.

This insight naturally gives rise to a compelling question regarding the future of linear attention: Does the Diagonal-Plus-Rank-1 decay structure truly represent the celling of hidden state management and utilization? Or to say, can we extend the decay matrices to broader, structured, and more expressive classes, thus further elevating the performance ceiling of linear attention mechanisms?

Our work addresses the aforementioned questions through the following two primary contributions.

Firstly, we propose the Householder-Diagonalized Linear Attention (HDLA) method, which augments language modeling capacity via a more sophisticated decay matrix structure while maintaining reasonable computational costs. We refine three necessary restrictions when designing complex and efficient decay structures: parameter efficiency, memory efficiency, and computational efficiency. Based on these restrictions and inspired by the congruence diagonalization theory of real symmetric matrices, we employ generalized Householder matrices to diagonalize the decay matrix, and show that HDLA's structured decay is a specific instance of the Diagonal-Plus-Rank-2 class.

Secondly, we introduce a rank-generalized chunk-wise parallel algorithmic framework, which simultaneously accommodates the arbitrary diagonal term in linear attention mechanisms' decay. When formulating the chunk-wise parallel algorithm for HDLA, we achieve a broad generalization that accommodates both Diagonal-Plus-Rank-$r_{ab}$ decay structures and rank-$r_{kv}$ key-value updates. This advance not only subsumes HDLA as a special case, but also provides a robust foundation for future research on linear attention mechanisms.

Comprehensive experiments fully demonstrate the superior performance of our proposed HDLA model: (1) Achieves state-of-the-art (SOTA) results in terms of language modeling perplexity (up to 2.8B parameter scale), with retrieval capability at 2.8B scale surpassing all linear attention baselines. (2) On the retrieval-based RULER (Hsieh et al., 2024) experiment, achieves up to a 58.2% accuracy improvement compared to Gated DeltaNet (Yang et al., 2025). (3) In synthetic MAD (Poli et al.) experiment, the average score exceeds linear attention baselines by 4.39-7.66, significantly narrowing the performance gap with Softmax Attention (Vaswani et al., 2017). (4) In synthetic MQAR (Arora et al., 2023a) experiments, at sequence length 2048, the accuracy is about 80% higher than the more computationally intensive Gated DeltaProduct with $n_h = 2$ (Siems et al., 2025).

While achieving superior performance, HDLA maintains a relatively limited and reasonable computation amount. Even when compared to Gated DeltaProduct with $n_h = 3$ (Siems et al., 2025), whose computation amount is about 2x that of HDLA, HDLA still shows a clear performance superiority.

Our HDLA model and the generalized chunk-wise parallel algorithm together provide a foundation for future research on rank-enhanced structured linear attention, showcasing promising prospects.

## 2 BACKGROUNDS AND RELATED WORKS

For notational conventions in this work, we use bold lowercase letters to denote column vectors (e.g., $\mathbf{q}_t$), bold uppercase letters for matrices (e.g., $\mathbf{Q}, \mathbf{O}$), and italic uppercase letters for learnable parameters (e.g., $\boldsymbol{\theta}_q$). Note that any matrix without a subscript is constructed by concatenating its corresponding lowercase column vectors, e.g., $\mathbf{Q}$ denotes the column-wise concatenation of $\mathbf{q}_1, \mathbf{q}_2, \cdots$. We also use lowercase letters to represent tensors of a single timestep with more than 1 columns. For instance, $\mathbf{k}_t \in \mathbb{R}^{d_k \times r_{kv}}$, and $\mathbf{K}$ is assembled by concatenating $\mathbf{k}_1, \mathbf{k}_2, \cdots$ column-wise.

**Unified Recurrent Form of Linear Attention.** In linear attention, an input $\mathbf{x}_t \in \mathbb{R}^{d \times 1}$ is transformed into a group of query $\mathbf{q}_t \in \mathbb{R}^{d_k \times 1}$, key $\mathbf{k}_t \in \mathbb{R}^{d_k \times r_{kv}}$ and value $\mathbf{v}_t \in \mathbb{R}^{d_v \times r_{kv}}$ at first:

$$\mathbf{q}_t = f_q(\mathbf{x}_t, \boldsymbol{\theta}_q), \mathbf{k}_t = f_k(\mathbf{x}_t, \boldsymbol{\theta}_k), \mathbf{v}_t = f_v(\mathbf{x}_t, \boldsymbol{\theta}_v) \tag{1}$$

The above transformation $f_q, f_k, f_v$ are typically linear functions, possibly with activation or normalization, and $\boldsymbol{\theta}_q, \boldsymbol{\theta}_k, \boldsymbol{\theta}_v$ are their projection parameters. Then, hidden state $\mathbf{S}_t \in \mathbb{R}^{d_k \times d_v}$, decay matrix $\mathbf{P}_t \in \mathbb{R}^{d_k \times d_k}$, and the attention output $\mathbf{o}_t \in \mathbb{R}^{d_v \times 1}$ are computed as follows:

$$\mathbf{P}_t = f_p(\mathbf{x}_t, \boldsymbol{\theta}_p) \in \mathbb{R}^{d_k \times d_k} \tag{2}$$

$$\mathbf{S}_t = \mathbf{P}_t \mathbf{S}_{t-1} + \mathbf{k}_t \mathbf{v}_t^\top \in \mathbb{R}^{d_k \times d_v}, \tag{3}$$

$$\mathbf{o}_t = \mathbf{S}_t^\top \mathbf{q}_t \in \mathbb{R}^{d_v \times 1} \tag{4}$$

The hidden state $\mathbf{S}_t$ seeks to compress information from arbitrarily long key-value pairs into a fixed-size memory. The decay matrix $\mathbf{P}_t$ balances the relative importance between historical information $\mathbf{S}_{t-1}$ and the incoming new information $\mathbf{k}_t \mathbf{v}_t^\top$. Different structures of $\mathbf{P}_t$ lead to different levels of model performance, as well as varying parallel forms and strategies of sequential parallelism.

**The original purpose of linear attention.** Linear attention (Katharopoulos et al., 2020) is originally targeted at addressing the time and space complexity issue of Softmax attention. It uses linear kernel functions to approximate the high-cost non-linear Softmax operation. Leveraging the associative property of matrix multiplication, it enables each key-value pair to be processed only once, and achieves $O(n)$ time complexity while compressing infinite key-value sequences into fixed size $\mathbf{S}_t$.

**The developmental trajectory of decay matrices.** The evolution of linear attention methods has moved from the original variant (Katharopoulos et al., 2020) lacking any decay mechanisms—which cannot forget unimportant historical information—to approaches with learnable constant decay such as RetNet (Sun et al., 2023) and TransNormer (Qin et al., 2024a). While these mitigate forgetting to a certain extent, they remain insensitive to the relative importance between historical information $\mathbf{S}_{t-1}$ and newly arriving information $\mathbf{k}_t \mathbf{v}_t^\top$. More recently, diagonal input-dependent decay mechanisms, introduced by models such as GLA (Yang et al., 2024a), Mamba (Gu & Dao, 2024), and HGRN2 (Qin et al., 2024d), enable adaptive weighting of historical context but are constrained by their diagonal structure, leading to a lack of cross-row interaction during hidden state updates. As a result, these mechanisms permit only partial forgetting of old information without negative erasure. To address this, recent works have adopted input-dependent non-diagonal decay structures (typically Diagonal-Plus-Rank-1) and have demonstrated superior performance over earlier approaches. DeltaNet (Yang et al., 2024b) and TTT-Linear (Sun et al., 2024) were the first to employ generalized Householder matrices as non-diagonal decay matrices. Gated DeltaNet further improves language modeling capabilities by incorporating a scalar forget gate into DeltaNet. RWKV-7 (Peng et al., 2025) adopts a more general diagonal-plus-rank-1 decay structure, in which the diagonal terms are analogous to the input-dependent decay used in GLA. Gated DeltaProduct (Siems et al., 2025) repeats the recurrent step of Gated DeltaNet for $n_h$ times at each timestep, which is equivalent to applying a Diagonal-Plus-Rank-$n_h$ single-step rank-enhanced decay. The resulting decay matrix lacks strong structural properties, and its performance improvement is limited even as the computational amount grows 1 or 2 times. Therefore, we aim to explore a structured rank-enhanced decay method that achieves greater performance gains, while incurring only limited additional computational cost compared to Diagonal-Plus-Rank-1.

**Chunk-wise parallel acceleration algorithm of linear attention.** The core idea of chunk-wise parallel algorithms for linear attention is to divide the computation along the time dimension into chunks, sequentially compute the checkpoints of hidden states before entering each sequential chunk, and then process the linear attention outputs of different time intervals in parallel. Lightning-

Table 1: **The structures of decay $\mathbf{P}_t$ in different linear attention mechanisms** ($\alpha, \alpha_t, \beta_t \in \mathbb{R}; \mathbf{k}_t, \boldsymbol{\lambda}_t, \mathbf{a}_t, \mathbf{w}_t, \hat{\kappa}_t \in \mathbb{R}^{d_k \times 1}$).

| Model | $\mathbf{P}_t$ |
|---|---|
| Original Linear Attention | $\mathbf{I}$ |
| RetNet, TransNormer | $\alpha \mathbf{I}$ |
| GLA, Mamba, HGRN2 | $\mathrm{Diag}(\boldsymbol{\lambda}_t)$ |
| DeltaNet, TTT-Linear | $\mathbf{I} - \beta_t \mathbf{k}_t \mathbf{k}_t^\top$ |
| Gated DeltaNet | $\alpha_t(\mathbf{I} - \beta_t \mathbf{k}_t \mathbf{k}_t^\top)$ |
| Gated DeltaProduct ($n_h$ iterations) | $\alpha_t(\mathbf{I} - \beta_t \mathbf{k}_t \mathbf{k}_t^\top)$ |
| RWKV-7 | $\mathrm{Diag}(\mathbf{w}_t) - \hat{\kappa}_t(\mathbf{a}_t \odot \hat{\kappa}_t)^\top$ |

Attention (Qin et al., 2024b) and Lightning-Attention-2 (Qin et al., 2024c) address the parallelization problem in the case of diagonal scalar decay, while Yang et al. (2024a) tackles the parallelization for diagonal vector decay. ZeCO (Chou et al., 2025) further addresses the communication bottleneck in multi-GPU scaling based on previous algorithms. (Gated) DeltaNet (Yang et al., 2024b; 2025) solves the parallelization for the case of diagonal plus rank-1 decay. ParallelFlow (Cirone & Salvi, 2025) provides a certain degree of parallelism for identity plus rank-$n$ decay, but it does not accommodate the arbitrary diagonal terms that are common in the decay matrices of linear attention.

**Test-Time training.** If the hidden state $\mathbf{S}_t$ is regarded as the projection parameter of a linear layer, then the autoregressive update formula for the hidden state in most linear attention mechanisms can be interpreted as stochastic gradient descent (SGD) on $\mathbf{S}_t$, usually aiming at next value prediction (using $\mathbf{k}_t^\top \mathbf{S}_t$ to predict $\mathbf{v}_t$). This update process is referred to as Test-Time Training. TTT-Linear (Sun et al., 2024) and DeltaNet (Yang et al., 2024b) were the first to interpret and design linear attention mechanisms from this perspective. Titans (Behrouz et al., 2025c) introduces momentum to the stochastic gradient descent. Miras (Behrouz et al., 2025b) proposes a broad unifying framework that integrates linear attention and Softmax attention under the view of Test-Time Training, utilizing components such as memory architectures, memory learning methods, attention bias, and retention gates. ATLAS (Behrouz et al., 2025a) and MesaNet (von Oswald et al., 2025) make improvements on the stochastic gradient descent (attention bias) objective by optimizing the average loss of all tokens within a sliding window or a global window, thereby achieving better performance.

## 3 METHOD

### 3.1 LINEAR ATTENTION WITH HOUSEHOLDER-DIAGONALIZED DECAY

Our goal is to achieve better language modeling capabilities through extending the decay matrices to a broader, structured, and more expressive class, while simultaneously meeting efficiency constraints in parameters, memory, and computation. Specifically, our idea is to parameterize the Diagonal-Plus-Rank-2 decay structure by utilizing a certain kind of efficient matrix decomposition method.

#### 3.1.1 EFFICIENCY CONSTRAINTS OF COMPLEX DECAY MATRIX DESIGN

Parameter, memory, and computational efficiency are common challenges during the design of linear attention mechanisms. When designing complex decay structures, we'd like to revisit and refine these constraints, so as to limit the extremely broad design space and to preliminarily validate the practicality of our approach. (1) Parameter efficiency. The $O(d_k^2)$ decay matrix should be obtained through $O(d_k)$ parameters, to maintain a balance with the parameter counts of $\boldsymbol{\theta}_Q, \boldsymbol{\theta}_K$ and $\boldsymbol{\theta}_V$, avoiding excessive parameters and learning overhead. (2) Memory efficiency. Each of the $O(d_k^2)$ decay matrices or their cumulative products should be compactly stored in $O(d_k)$ memory on average,

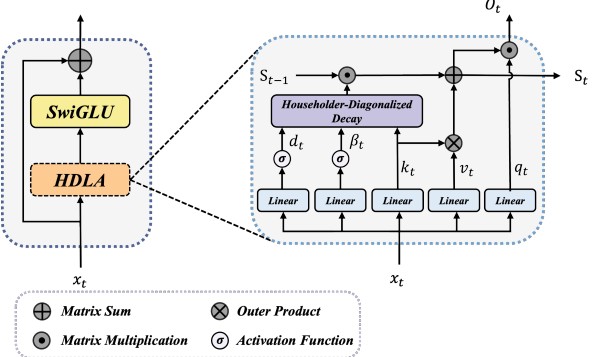

Figure 1: **The architecture of HDLA, as well as its integration within a Transformer layer**. Details like output gates and activation on keys and values are omitted. See appendix C.1 for the exact architecture of HDLA.

matching the memory footprint of $\mathbf{q}_t, \mathbf{k}_t$ and $\mathbf{v}_t$. (3) Computational efficiency. The cumulative product of decay matrices must maintain reasonable computational costs. Moreover, hidden state updates across sequential blocks should be enabled through concise one-pass matrix multiplications.

#### 3.1.2 HOUSEHOLDER DIAGONALIZED LINEAR ATTENTION (HDLA)

To efficiently parameterize a complex decay $\mathbf{P}_t$, it is advantageous to decompose it into simpler components through matrix decomposition theory. Note that any real symmetric matrix $\mathbf{P}_t$ can undergo congruence diagonalization via some invertible matrix $\mathbf{H}_t \in \mathbb{R}^{d_k \times d_k}$, i.e., $\mathbf{P}_t = \mathbf{H}_t \boldsymbol{\Lambda}_t \mathbf{H}_t^\top$. Utilizing this inspiration, the parameterization of $\mathbf{P}_t$ can be reduced to two sub-problems: (P1) Learning the diagonal eigenvalue matrix $\boldsymbol{\Lambda}_t$. (P2) Selecion of the invertible transformation $\mathbf{H}_t$.

For (P1), we make the parameterization of $\boldsymbol{\Lambda}_t$ analogous to GLA's input-dependent diagonalized decay, equipping the model with fundamental capability to dynamically forget historical information. For (P2), we adopt generalized Householder matrices as our transformation operator, inspired by recent research of Diagonal-Plus-Rank-1 decay structure (Yang et al., 2024b; Sun et al., 2024; Yang et al., 2025; Siems et al., 2025). The corresponding hidden state update formulae are as follows:

$$\mathbf{P}_t = (\mathbf{I} - \beta_t \mathbf{k}_t \mathbf{k}_t^\top) \mathbf{\Lambda}_t (\mathbf{I} - \beta_t \mathbf{k}_t \mathbf{k}_t^\top) \in \mathbb{R}^{d_k \times d_k}, \tag{5}$$

$$\mathbf{\Lambda}_t = \mathrm{Diag}(\lambda_t) \in \mathbb{R}^{d_k \times d_k}, \lambda_t = \sigma(\boldsymbol{W}_\Lambda \mathbf{x}_t) \in \mathbb{R}^{d_k \times 1}, \tag{6}$$

We make $\beta_t \in (0, 2)$ to enhance the model's state tracking capability, following the conclusion of Grazzi et al. (2025). $\sigma(\cdot)$ is an activation function ranging in $(0, 1)$, and we adopt $\mathrm{sigmoid}(\cdot)$ here.

Compared with GLA, the only excessive parameter is the projection matrix (of $O(d_k)$ scale) mapping the input $\mathbf{x}_t$ into $\beta_t$, confirming the parameter efficiency of HDLA. Deduction of chunk-wise parallel algorithm in the following section will verify its computational and memory efficiency.

## 3.2 GENERALIZED CHUNK-WISE PARALLEL ALGORITHM

### 3.2.1 DERIVATION AND RANK EXTENSION OF A GENERALIZED HIDDEN STATE UPDATE RULE

During the derivation of HDLA's chunk-wise parallel algorithm, we first reformulate its decay matrix as a special case of the Diagonal-Plus-Rank-2 structure (see Appendix appendix C.2 for details):

$$\mathbf{P}_t = \mathbf{D}_t - \mathbf{A}_t \mathbf{B}_t^\top \in \mathbb{R}^{d_k \times d_k}, \mathbf{A}_t, \mathbf{B}_t \in \mathbb{R}^{d_k \times 2} \tag{7}$$

Based on the above reformulation, and to provide a foundational support for future research both theoretically and practically, we aim to develop a broader chunk-wise parallel algorithm for the following hidden state recurrent update rule, which generalizes the ranks of $\mathbf{A}_t \mathbf{B}_t^\top$, and $\mathbf{K}_t \mathbf{V}_t^\top$ to arbitrary values simultaneously (i.e., setting $\mathbf{A}_t, \mathbf{B}_t \in \mathbb{R}^{d_k \times r_{ab}}, \mathbf{K}_t \in \mathbb{R}^{d_k \times r_{kv}}, \mathbf{V}_t \in \mathbb{R}^{d_v \times r_{kv}}$):

$$\mathbf{S}_t = (\mathbf{D}_t - \mathbf{A}_t \mathbf{B}_t^\top) \mathbf{S}_{t-1} + \mathbf{K}_t \mathbf{V}_t^\top \in \mathbb{R}^{d_k \times d_v} \tag{8}$$

### 3.2.2 RANK GENERALIZED CHUNK-WISE PARALLEL ALGORITHM

**Notational conventions.** Define two kinds of matrices' cumulative products as follows:

$$\mathbf{P}_i^j = \begin{cases} \prod_{t=i+1}^j \mathbf{P}_t, & i < j \\ \mathbf{I}, & i \geq j \end{cases}, \mathbf{D}_i^j = \begin{cases} \prod_{t=i+1}^j \mathbf{D}_i, & i < j \\ \mathbf{I}, & i \geq j \end{cases}, \mathbf{d}_i^j = \mathbf{D}_i^j \mathbf{1} \in \mathbb{R}^{d_k \times 1} \tag{9}$$

All timesteps in this work start at 1. The input tensors are partitioned along the sequential dimension into chunks of size $C$. We abuse the subscript $[n]$ to refer to tensors relevant to the $n$-th sequential chunk. $\mathbf{A}_{[n]} \in \mathbb{R}^{d_k \times r_{ab} C}, \mathbf{B}_{[n]} \in \mathbb{R}^{d_k \times r_{ab} C}, \mathbf{K}_{[n]} \in \mathbb{R}^{d_k \times r_{kv} C}, \mathbf{V}_{[n]} \in \mathbb{R}^{d_v \times r_{kv} C}$ are concatenated column-wise from the corresponding input tensors of each timestep inside the chunk, while $\mathbf{S}_{[n]} = \mathbf{S}_{(n-1)C}$ denotes the hidden state right before processing the first timestep of the $n$-th chunk.

**Computation Flow.** Since the linear attention of $\mathbf{q}_{nC+t}$ over the first $nC$ tokens can be coalesced into the interaction between $\mathbf{q}_{nC+t}, \mathbf{S}_{[n]}$, and $\mathbf{P}_{nC}^{nC+t}$, our method adopts a two-phase computation scheme similar to Lightning Attention (Qin et al., 2024b) and Gated Linear Attention (Yang et al., 2024a): (1) Sequentially computing the hidden state checkpoints $\mathbf{S}_{[0]}, \mathbf{S}_{[1]}, ..., \mathbf{S}_{[N-1]}$; and (2) Computing the linear attention outputs $\mathbf{O}_{[0]}, ..., \mathbf{O}_{[N-1]}$ across different time ranges in parallel.

These two computation phases correspond to the following eq. (10) and eq. (11), respectively:

$$\mathbf{S}_{[n]} = \mathbf{P}_{(n-1)C}^{nC} \mathbf{S}_{[n-1]} + \sum_{t=(n-1)C+1}^{nC} \mathbf{P}_t^{nC} \mathbf{K}_t \mathbf{V}_t^\top \in \mathbb{R}^{d_k \times d_v}, \tag{10}$$

$$\mathbf{o}_{(n-1)C+t} = \underbrace{\mathbf{S}_{[n-1]}^\top \mathbf{P}_{(n-1)C}^{(n-1)C+t} \mathbf{q}_{(n-1)C+t}}_{inter-chunk\ attention} + \big( \underbrace{\sum_{i=(n-1)C+1}^{(n-1)C+t} \mathbf{V}_i \mathbf{K}_i^\top \mathbf{P}_t^{(n-1)C+t}}_{intra-chunk\ attention} \big) \mathbf{q}_t \tag{11}$$

**Rank Generalized WY Representation.** Let $\mathbf{P}_{[n]} = \mathbf{P}_{(n-1)C}^{nC} = \prod_{t=(n-1)C}^{nC} \mathbf{P}_t \in \mathbb{R}^{d_k \times d_k}$, $\mathbf{H}_{[n]} = \sum_{t=(n-1)C+1}^{nC} \mathbf{P}_t^{nC} \mathbf{K}_t \mathbf{V}_t^\top \in \mathbb{R}^{d_k \times d_v}$. For efficient computation of eq. (10) and eq. (11), it becomes imperative to identify optimized representations for both $\mathbf{P}_{[n]}$ and $\mathbf{H}_{[n]}$ that eliminate their original dependence on cumulative summation ($\Sigma$) and cumulative product ($\prod$) operators.

Employing mathematical induction, we optimize the representations of $\mathbf{P}_{[n]}$ and $\mathbf{H}_{[n]}$ as follows:

$$\mathbf{P}_{[n]} = \mathbf{D}_{(n-1)C}^{nC}(\mathbf{I} - \mathbf{B}_{[n]}'\mathbf{W}_{[n]}^\top), \mathbf{H}_{[n]} = \mathbf{D}_{(n-1)C}^{nC}(\mathbf{K}_{[n]}'\mathbf{V}_{[n]}^\top - \mathbf{B}_{[n]}'\mathbf{U}_{[n]}^\top) \tag{12}$$

$\mathbf{U}_{[n]} \in \mathbb{R}^{d_v \times r_{ab}C}$ and $\mathbf{W}_{[n]} \in \mathbb{R}^{d_k \times r_{ab}C}$ are core components of arbitrary rank WY representation:

$$\mathbf{U}_{[n]} = \mathbf{V}_{[n]}\text{triu}_{r_{kv} \times r_{ab}}(\mathbf{K}_{[n]}'\mathbf{A}_{[n]}'^\top, 1)\left(\mathbf{I} + \text{triu}_{r_{ab} \times r_{ab}}(\mathbf{B}_{[n]}'\mathbf{A}_{[n]}'^\top, 1)\right)^{-1} \in \mathbb{R}^{d_v \times r_{ab}C}, \tag{13}$$

$$\mathbf{W}_{[n]} = \mathbf{A}_{[n]}'\left(\mathbf{I} + \text{triu}_{r_{ab} \times r_{ab}}(\mathbf{B}_{[n]}'\mathbf{A}_{[n]}'^\top, 1)\right)^{-1} \in \mathbb{R}^{d_k \times r_{ab}C}, \tag{14}$$

The above custom operator $\text{triu}_{r_1 \times r_2}(\mathbf{R}_{r_1 \times n, r_2 \times n}, i)$ serves analogous to standard $\text{triu}(\mathbf{R}_{n \times n}, i)$ in linear attention, except for treating each $r_1 \times r_2$ sub-block of $\mathbf{R}_{r_1 \times n, r_2 \times n}$ as a single element.

$\mathbf{A}_{[n]}'$ is obtained from $\mathbf{A}_{[n]}$ using the following element-wise multiplication ($\odot$) on each of its column vectors (e.g., $\mathbf{A}_{[n],:,t\cdot r_{ab}+r}'$), where $t$ is the time index inside the sequential chunk, and $r$ is the rank index. $\mathbf{B}_{[n]}'$ and $\mathbf{K}_{[n]}'$ are obtained similar to $\mathbf{A}_{[n]}'$, but with element-wise division ($\oslash$) instead:

$$\mathbf{A}_{[n],:,t\cdot r_{ab}+r}' = \mathbf{A}_{[n],:,t\cdot r_{ab}+r} \odot \mathbf{d}_{(n-1)C}^{(n-1)C+(t-1)} \tag{15}$$

$$\mathbf{B}_{[n],:,t\cdot r_{ab}+r}' = \mathbf{B}_{[n],:,t\cdot r_{ab}+r} \oslash \mathbf{d}_{(n-1)C}^{(n-1)C+t}, \mathbf{K}_{[n],:,t\cdot r_{kv}+r}' = \mathbf{K}_{[n],:,t\cdot r_{kv}+r} \oslash \mathbf{d}_{(n-1)C}^{(n-1)C+t} \tag{16}$$

**Resulting Formulae.** Leveraging the WY representation defined in eq. (13) and eq. (14), eq. (10) and eq. (11) can be reformulated into the following form, enabling efficient parallel computation of attention output in different time range, after sequential computation of hidden state checkpoints:

$$\mathbf{S}_{[n]} = \mathbf{D}_{(n-1)C}^{nC}(\mathbf{I} - \mathbf{B}_{[n]}'\mathbf{W}_{[n]}^\top)\mathbf{S}_{[n-1]} + \mathbf{D}_{(n-1)C}^{nC}(\mathbf{K}_{[n]}'\mathbf{V}_{[n]}^\top - \mathbf{B}_{[n]}'\mathbf{U}_{[n]}^\top) \tag{17}$$

$$\mathbf{O}_{[n]} = \mathbf{S}_{[n-1]}^\top \mathbf{Q}_{[n]}' + \mathbf{V}_{[n]}\text{triu}_{r_{kv} \times 1}(\mathbf{K}_{[n]}'^\top\mathbf{Q}_{[n]}', 0) - (\mathbf{S}_{[n-1]}^\top\mathbf{W}_{[n]} + \mathbf{U}_{[n]})\text{triu}_{r_{ab} \times 1}(\mathbf{B}_{[n]}'^\top\mathbf{Q}_{[n]}', 0), \tag{18}$$

Here we only present some key conclusions. For detailed derivations, please refer to appendix C.3.

## 3.3 DISCUSSIONS

**Understanding HDLA from the Perspective of Test-Time Training (TTT).** If $\mathbf{S}_t$ is regarded as the projection parameter of a linear layer, then a single step of hidden state update in HDLA is equivalent to the following three-step optimization process (see appendix C.5 for details):

$$\mathbf{S}_{t,1} = \mathbf{S}_{t-1} - \frac{\beta_t}{2}\nabla(\min_{S_{t-1}} \|\mathbf{k}_t^\top \mathbf{S}_{t-1}\|^2), \tag{19}$$

$$\mathbf{S}_{t,2} = \mathbf{S}_{t,1} - \frac{1}{2}\nabla\left(\text{Trace}(\mathbf{S}_{t,1}^\top\text{diag}(1 - \lambda_t)\mathbf{S}_{t,1})\right), \tag{20}$$

$$\mathbf{S}_t = \mathbf{S}_{t,2} - \frac{\beta_t}{2}\nabla(\mathbf{S}_{t,2}\|\mathbf{k}_t^\top \mathbf{S}_{t,2} - \frac{1}{\beta_t}\mathbf{v}_t^\top\|^2). \tag{21}$$

HDLA is characterized by preemptively penalizing high inner product similarity between $\mathbf{k}_t$ and each column of $\mathbf{S}_{t-1}$ (as demonstrated in eq. (19)), which partially removes redundant information stored in $\mathbf{S}_{t-1}$ before performing gradient descent using delta rule (Yang et al., 2024b) in eq. (21).

**Comparisons between HDLA v.s. Gated DeltaProduct.** Gated DeltaProduct performs $n_h$ value predictions and optimizations at each timestep. According to Yang et al. (2024b), all its iterations within a single timestep can be merged into a rank-enhanced iteration with $r_{ab} = r_{kv} = n_h$. However, its coalesced Diagonal-Plus-Rank-$r_{ab}$ decay does not exhibit a highly structured pattern. We will demonstrate in the experiments that even when $n_h = 3$, the computation amount is about $2\times$ of HDLA, the performance of Gated DeltaProduct still falls considerably short of our method.

Table 2: **Comparison on the computation amounts of HDLA, GDP2 (Gated DeltaProduct, $n_h = 2$), and GDP3 (Gated DeltaProduct, $n_h = 3$) of a single recurrent timestep.** We uniformly calculate the computational cost of recurrent hidden state updates according to eq. (17) (setting $C = 1$), and omit the estimation of computation required by the cumbersome WY Representation.

| Method | $r_{ab}$ | $r_{kv}$ | Input Projection | Hidden State Update | Output Generation |
|--------|------|------|------------------|---------------------|-------------------|
| HDLA | 2 | 1 | $d(3d_k + d_v + 1)$ | $d_k(8d_v + 5)$ | $d_k d_v$ |
| GDP2 | 2 | 2 | $d(3d_k + 2d_v + 3)$ | $d_k(12d_v + 6)$ | $d_k d_v$ |
| GDP3 | 3 | 3 | $d(4d_k + 3d_v + 4)$ | $d_k(18d_v + 9)$ | $d_k d_v$ |

## 4 EXPERIMENTS

We've conducted a series of experiments, ranging from synthetic tasks (MAD and Zoology), language modeling experiments, retrieval-based tasks (NIAH), image classification and ablation studies, to comprehensively validate the effectiveness of our model. In the following, we use GDP2 and GDP3 as abbreviations for Gated DeltaProduct when $n_h = 2$ and $n_h = 3$, respectively. Both models incur significantly higher computational and memory overhead compared to HDLA, yet their overall performance still remains inferior to our proposed method. (See appendix D for detailed settings)

**Mechanistic Architectural Design (MAD).** The MAD benchmark (Poli et al.) is composed of 6 kinds of small-scale synthetic tasks, and is designed to evaluate a model's core language modeling capabilities including in-context recall, memorization, information compression, selective copying and noise suppression, etc. The scores across all synthetic tasks are averaged to predict the model's performance at large scales, according to scaling law (Kaplan et al., 2020; Shen et al., 2024).

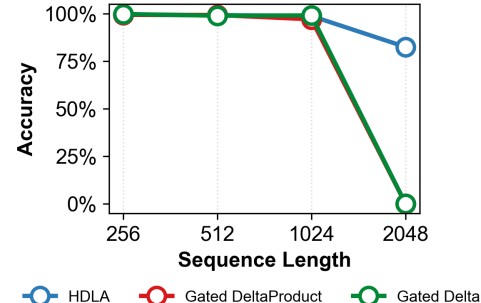

Figure 2: **Accuracy on the synthetic MQAR task.**

Table 3: **Performance Comparison on MAD benchmark aligned with MAD protocol** Mem: Memorization. ICR: In-Context Recall

| Method | Compression | Fuzzy ICR | ICR | Mem. | Noisy ICR | Selective Copy | AVG. |
|--------|-------------|-----------|-----|------|-----------|----------------|------|
| Softmax Attention | 48.85 | **39.74** | 95.98 | 84.41 | 88.12 | 99.03 | **76.02** |
| GDP2 | 39.40 | 10.59 | 99.29 | 49.84 | 95.06 | 97.68 | 65.31 |
| DeltaProduct | 40.77 | 14.16 | 99.85 | 46.08 | 99.66 | **99.95** | 66.74 |
| Gated DeltaNet | 41.41 | 12.90 | 99.73 | 55.64 | 99.40 | 99.91 | 68.17 |
| DeltaNet | 42.27 | 16.42 | **99.88** | 42.46 | **99.85** | 99.93 | 66.80 |
| Mamba | 48.20 | 10.24 | 86.90 | **89.48** | 94.50 | 82.14 | 68.58 |
| **HDLA** | **51.01** | 14.56 | 99.73 | 89.34 | 93.42 | 89.73 | 72.97 |

The results in table 3 demonstrate that: (1) The 4 non-diagonal decay baselines suffer from severely impaired memorization capability, with scores not exceeding 60, whereas our HDLA performs well. (2) Compared to the five linear attention baselines, HDLA demonstrates balanced and comprehensive advantages across all tasks, and significantly narrows the performance gap with softmax attention. (3) HDLA underperforms softmax attention on Fuzzy In-Context Recall – a task requiring accurate value prediction from keys interleaved wth arbitrary noisy tokens. This kind of limitation

can be attributed to the strong recency bias (Pan et al., 2025) in linear attention mechanisms - where the diagonal decay coefficients decrease cumulatively over time. As a result, HDLA cannot achieve a high upper bound for attention scores on distant tokens, thereby failing to effectively aggregate information from important tokens distributed sparsely and discretely, and ultimately impairing its retrieval capability.

**Multi-Query Associative Recall (Zoology).** We conduct Multi-Query Associative Recall (Zoology, Arora et al. (2023a)) experiment against Gated DeltaProduct ($n_h = 2$) (Siems et al., 2025) and Gated DeltaNet (Yang et al., 2025), with parameter scale aligned to 1.65M. The evaluated lengths include 256, 512, 1024 and 2048. See fig. 2 for the results. When the maximum evaluation length is extended to 2048, HDLA still maintains an accuracy higher than 81%, while the two baselines nearly fail to produce correct answers, demonstrating HDLA's advantage in recall ability.

Table 4: **Perplexity comparison on language modeling.** The parameter scales of the three columns from left to right are: 0.4B, 1.45B and 2.8B, respectively. Wiki: Wikitext. (Merity et al., 2016) LMB: Lambada (Paperno et al., 2016).

| Model | Wiki ppl ↓ | LMB. ppl ↓ | Avg. ppl ↓ | Wiki ppl ↓ | LMB. ppl ↓ | Avg. ppl ↓ | Wiki ppl ↓ | LMB. ppl ↓ | Avg. ppl ↓ |
|---|---|---|---|---|---|---|---|---|---|
| **Linear Attention** | | | | | | | | | |
| **HDLA** | 29.04 | **43.09** | **36.06** | 22.49 | **22.16** | **22.32** | **20.16** | **16.99** | **18.58** |
| GDP2 (Siems et al., 2025) | 30.98 | 51.59 | 41.28 | 23.51 | 25.79 | 24.65 | 20.94 | 19.82 | 20.38 |
| GDP3 (Siems et al., 2025) | 31.52 | 60.92 | 46.22 | 24.63 | 28.97 | 26.80 | - | - | - |
| Gated DeltaNet (Yang et al., 2025) | 30.06 | 56.07 | 43.06 | 23.09 | 26.56 | 24.83 | 20.47 | 18.74 | 19.60 |
| DeltaNet (Yang et al., 2024b) | 30.75 | 58.34 | 44.54 | 23.74 | 31.14 | 27.44 | 21.66 | 23.72 | 22.69 |
| HGRN2 (Qin et al., 2024d) | 30.87 | 47.81 | 39.34 | 23.26 | 24.70 | 23.98 | 20.93 | 19.69 | 20.31 |
| Mamba2 (Dao & Gu, 2024) | 30.26 | 51.00 | 40.63 | 23.93 | 27.53 | 25.73 | 21.95 | 23.61 | 22.78 |
| GLA (Yang et al., 2024a) | 30.95 | 56.55 | 43.75 | 23.44 | 29.41 | 26.42 | 21.08 | 21.82 | 21.45 |
| TransNormerLLM (Qin et al., 2024a) | 31.33 | 51.17 | 41.25 | 24.15 | 28.41 | 26.28 | 21.47 | 21.97 | 21.72 |
| **Softmax-Attention** | | | | | | | | | |
| Llama (Touvron et al., 2023) | **28.46** | 46.73 | 37.60 | **22.29** | 25.07 | 23.68 | 20.32 | 21.10 | 20.71 |

**Language Modeling.** We train 3 parameter scales of all the models: 0.4B, 1.45B and 2.8B on 10B/50B token datasets sampled from FineWeb-Edu. Perplexity results in table 4 demonstrate that HDLA surpasses all the selected linear attention baselines by notable margins, and even outperforms the Transformer-based architecture Llama (Touvron et al., 2023). table 7 shows that our method consistently surpasses both Llama and linear attention baselines in zero-shot commonsense reasoning. For retrieval tasks, our method is competitive in all parameter scales, and achieves the best performance among all linear attention mechanisms when scaled up to 2.8B parameters. However, there is still a considerable gap between our model and Llama in retrieval performance. The reason is that the limited hidden state size of linear attention mechanisms fundamentally restricts their ability to perform in-context cross-step retrieval, both explicitly and implicitly (Wen et al., 2025).

**Retrieval-Based Tasks.** We further trained Gated DeltaNet and HDLA models with 1.45B parameters until the total number of tokens reached 50B (see table 7 for language modeling and retrieval results), and then evaluated the models on the retrieval-based task RULER (Hsieh et al., 2024). As demonstrated by table 6, compared to Gated DeltaNet, HDLA has a significant advantage in retrieval capability. Especially on the S-NIAH-3 task, its accuracy leads by 31.4% and 58.2%.

Table 5: **Comparison on zero-shot commonsense reasoning and retrieval augmented generation with 50B training tokens.** We evaluate the models on BQ Clark et al. (2019), PIQA: Bisk et al. (2020), HS Zellers et al. (2019), WG Sakaguchi et al. (2021), Arc-e and Arc-c Clark et al. (2018), OBQ Mihaylov et al. (2018), SIQA Sap et al. (2019), SWDE Lockard et al. (2019), SC Rajpurkar et al. (2018) and FDA Arora et al. (2023b). AVG-CSR: Average CommonSense Reasoning accuracy. AVG-RET: Average RETrieval accuracy.

| Model | BQ. acc ↑ | PIQA acc ↑ | HS. acc-n ↑ | WG. acc ↑ | Arc-e acc ↑ | Arc-c acc-n ↑ | OBQ acc ↑ | SIQA acc ↑ | SWDE acc ↑ | SC acc ↑ | FDA acc ↑ | AVG-CSR acc ↑ | AVG-RET acc-n ↑ |
|---|---|---|---|---|---|---|---|---|---|---|---|---|---|
| **Parameter Scale: 1.45B, Number of tokens: 50B** | | | | | | | | | | | | | |
| **HDLA** | 1.45 | 73.50 | 57.33 | 57.62 | 73.44 | 38.14 | 41.60 | 42.02 | 41.40 | 36.76 | 16.61 | **54.81** | **31.59** |
| Gated DeltaNet | 1.45 | 73.23 | 56.23 | 56.51 | 72.43 | 38.14 | 41.20 | 39.71 | 37.89 | 35.86 | 16.88 | 53.92 | 30.21 |

Table 7: **Comparison on zero-shot commonsense reasoning and retrieval augmented generation with 10B training tokens.** We evaluate the models on BQ Clark et al. (2019), PIQA: Bisk et al. (2020), HS Zellers et al. (2019), WG Sakaguchi et al. (2021), Arc-e and Arc-c Clark et al. (2018), OBQ Mihaylov et al. (2018), SIQA Sap et al. (2019), SWDE Lockard et al. (2019), SC Rajpurkar et al. (2018) and FDA Arora et al. (2023b). AVG-CSR: Average CommonSense Reasoning accuracy. AVG-RET: Average RETrieval accuracy.

| Model | BQ. acc ↑ | PIQA acc ↑ | HS. acc-n ↑ | WG. acc ↑ | Arc-e acc ↑ | Arc-c acc-n ↑ | OBQ acc ↑ | SIQA acc ↑ | SWDE acc ↑ | SC acc ↑ | FDA acc ↑ | AVG-CSR acc ↑ | AVG-RET acc-n ↑ |
|---|---|---|---|---|---|---|---|---|---|---|---|---|---|
| **Parameter Scale: 0.4B, Number of tokens: 10B** | | | | | | | | | | | | | |
| **HDLA** | 61.50 | 67.41 | 40.48 | 51.14 | 60.65 | 28.58 | 31.80 | 38.13 | 10.26 | 21.78 | 3.09 | **47.46** | 11.71 |
| GDP2 | 61.07 | 66.87 | 38.49 | 57.70 | 57.70 | 27.73 | 34.00 | 38.08 | 9.36 | 22.39 | 3.63 | 46.97 | 11.79 |
| GDP3 | 60.37 | 66.59 | 37.62 | 51.30 | 57.53 | 26.28 | 35.00 | 38.84 | 8.37 | 20.68 | 3.36 | 46.69 | 10.80 |
| Gated DeltaNet | 58.41 | 67.63 | 39.41 | 51.85 | 58.38 | 27.13 | 33.60 | 36.75 | 8.01 | 20.78 | 2.63 | 46.65 | 10.47 |
| DeltaNet | 59.69 | 66.59 | 37.74 | 50.67 | 58.00 | 27.99 | 32.60 | 37.41 | 11.79 | 22.62 | 5.54 | 46.33 | 13.32 |
| HGRN2 | 59.17 | 67.08 | 38.96 | 52.09 | 60.02 | 26.62 | 34.80 | 38.43 | 9.90 | 18.83 | 3.45 | 47.15 | 10.73 |
| Mamba2 | 60.00 | 65.94 | 38.24 | 50.99 | 56.90 | 27.99 | 31.40 | 38.38 | 13.23 | 27.92 | 4.99 | 46.23 | 15.38 |
| GLA | 58.53 | 67.41 | 39.50 | 50.91 | 59.97 | 27.30 | 34.60 | 38.38 | 7.29 | 17.46 | 2.18 | 47.08 | 8.98 |
| TransNormerLLM | 59.45 | 66.59 | 38.34 | 49.64 | 59.51 | 28.41 | 35.60 | 39.56 | 10.08 | 21.31 | 2.00 | 47.14 | 11.13 |
| Llama | 60.73 | 66.65 | 38.88 | 51.62 | 58.63 | 28.24 | 33.40 | 38.95 | 47.07 | 30.86 | 17.15 | 47.14 | **31.69** |
| **Parameter Scale: 1.45B, Number of tokens: 10B** | | | | | | | | | | | | | |
| **HDLA** | 60.52 | 71.00 | 47.77 | 52.88 | 67.17 | 32.68 | 35.60 | 40.84 | 21.69 | 28.22 | 8.17 | **51.06** | 19.36 |
| GDP2 | 57.83 | 69.75 | 46.22 | 52.33 | 64.35 | 31.91 | 35.60 | 38.89 | 17.82 | 27.98 | 6.99 | 49.61 | 17.60 |
| GDP3 | 60.80 | 68.50 | 44.18 | 51.70 | 63.43 | 31.48 | 35.60 | 38.89 | 14.04 | 26.34 | 5.26 | 49.32 | 15.21 |
| Gated DeltaNet | 61.47 | 69.97 | 47.11 | 53.12 | 65.36 | 33.11 | 35.40 | 40.84 | 20.43 | 27.61 | 7.35 | 50.80 | 18.46 |
| DeltaNet | 61.31 | 69.31 | 44.32 | 53.04 | 65.32 | 31.23 | 34.80 | 39.61 | 21.87 | 26.91 | 10.25 | 49.87 | 19.68 |
| HGRN2 | 60.70 | 69.42 | 46.62 | 51.14 | 66.33 | 30.80 | 36.80 | 40.43 | 22.77 | 25.77 | 6.62 | 50.28 | 18.39 |
| Mamba2 | 60.46 | 69.70 | 45.00 | 51.78 | 63.43 | 31.23 | 34.60 | 39.87 | 22.23 | 29.42 | 9.35 | 49.51 | 20.33 |
| GLA | 57.31 | 69.31 | 47.25 | 54.06 | 66.46 | 33.79 | 36.60 | 39.82 | 16.29 | 23.83 | 4.81 | 50.58 | 14.98 |
| TransNormerLLM | 61.56 | 69.75 | 46.02 | 51.70 | 64.86 | 31.57 | 34.40 | 39.61 | 18.99 | 26.51 | 4.26 | 49.93 | 16.59 |
| Llama | 61.68 | 69.42 | 46.89 | 53.20 | 65.82 | 30.89 | 35.40 | 39.82 | 62.29 | 38.47 | 39.38 | 50.39 | **46.71** |
| **Parameter Scale: 2.8B, Number of tokens: 10B** | | | | | | | | | | | | | |
| **HDLA** | 61.13 | 71.65 | 51.93 | 56.51 | 70.29 | 34.90 | 37.60 | 40.69 | 27.45 | 30.56 | 16.61 | **53.09** | 24.87 |
| GDP2 | 58.75 | 71.16 | 50.31 | 55.41 | 67.59 | 34.73 | 38.40 | 40.17 | 27.00 | 30.56 | 8.71 | 52.07 | 22.09 |
| Gated DeltaNet | 60.80 | 71.76 | 51.17 | 54.54 | 69.49 | 35.67 | 38.20 | 40.43 | 29.07 | 31.13 | 13.79 | 52.67 | 24.66 |
| DeltaNet | 59.97 | 71.16 | 47.79 | 55.33 | 67.13 | 33.53 | 35.80 | 39.92 | 30.51 | 29.12 | 12.25 | 51.33 | 23.96 |
| HGRN2 | 61.56 | 70.57 | 50.49 | 53.04 | 68.90 | 34.81 | 39.00 | 40.43 | 28.44 | 29.19 | 14.61 | 52.35 | 24.08 |
| Mamba2 | 60.73 | 71.06 | 48.55 | 53.43 | 64.77 | 32.17 | 38.20 | 39.15 | 23.94 | 34.55 | 8.98 | 51.01 | 22.49 |
| TransNormerLLM | 58.59 | 70.29 | 50.04 | 54.54 | 68.35 | 33.96 | 35.60 | 41.76 | 24.21 | 29.42 | 7.62 | 51.64 | 20.42 |
| Llama | 61.10 | 70.89 | 50.36 | 56.20 | 67.38 | 32.51 | 36.20 | 40.07 | 61.57 | 36.23 | 41.02 | 51.84 | **46.27** |

Table 6: **Accuracy on different S-NIAH tasks for 1.45B HDLA and Gated DeltaNet.**

| Model | S-NIAH-1 | | S-NIAH-2 | | S-NIAH-3 | |
| Sequence Length | 1024 | 2048 | 1024 | 2048 | 1024 | 2048 |
|---|---|---|---|---|---|---|
| HDLA | **100.0%** | **98.8%** | **96.4%** | **52.2%** | **82.0%** | **65.2%** |
| Gated DeltaNet | 99.6% | 97.2% | **96.4%** | 45.8% | 50.6% | 7.0% |

**Image Classification.** We conduct bidirectional image classification experiments on ImageNet-1k (Deng et al., 2009). Baselines include Deit (Touvron et al., 2021) which is a Transformer-based architecture, and some other linear attention architectures. Results of baselines are directly borrowed from Chou et al. (2024). As show in table 8, HDLA performs better than most of the baselines.

**Supplementary Experiments**. In appendix B, we provide the following supplementary experiments: (1) State expansion experiments on HDLA and baselines. (2) Fine-tuning on some hyperparameters of HDLA (e.g. learning rate, the range of $\beta_t$, the type of activation functions on $\mathbf{k}_t, \mathbf{v}_t$).

Table 8: **Results of image classification on ImageNet-1k.**

| Model | Accuracy | Param(M) | Model | Accuracy | Param(M) | Model | Accuracy | Param(M) |
|---|---|---|---|---|---|---|---|---|
| HDLA | 74.84% | 6.1 | MetaLA | **75.33%** | 6.1 | GDP2 | 73.81% | 6.1 |
| Gated DeltaNet | 72.33% | 6.1 | HGRN | 74.40% | 6.1 | GLA | 72.47% | 6.1 |
| Mamba | 73.39% | 6.1 | Deit | 72.20% | 5.7 | - | - | - |

**Throughput comparison.** The training and inference throughput results for each model are presented in table 9, where the inference throughput is compared only during the decode stage. We acknowledge that the throughput of HDLA is lower than all baselines we've adopted. The reason is, compared to Gated DeltaNet, the vector decay, the number of key-value pairs, as well as the larger computational scale of WY Representation, all introduce significant additional overhead. In addition, we found that the mismatch between $r_{ab}$ and $r_{kv}$ also hinders kernel speed optimization.

Table 9: Training throughput and memory cost comparison between HDLA and baselines. All models have parameter scales aligned to 1.3B and are tested on a single A100-SXM-80GB GPU. Throughput is measured in Tokens Per Second (TPS), and Memory Cost is measured in GB.

| Model | Training | | Inference | |
|---|---|---|---|---|
| | Throughput (TPS) | Memory Cost (GB) | Throughput (TPS) | Memory Cost (GB) |
| GDN | 19277.18 | 39.33 | 15.3 | 5.3 |
| DP2 | 16335.64 | 46.58 | 14.4 | 5.3 |
| DP3 | 13492.85 | 54.99 | 14.9 | 5.3 |
| DeltaNet | 22060.12 | 39.74 | 14.7 | 5.3 |
| GDP2 | 14546.89 | 46.61 | 13.9 | 5.3 |
| GDP3 | 11769.86 | 55.06 | 13.7 | 5.3 |
| HGRN2 | 19229.97 | 46.22 | 14.3 | 5.2 |
| GLA | 20998.87 | 36.30 | 14.7 | 5.2 |
| HDLA | 7340.01 | 52.56 | 13.1 | 5.3 |

Nevertheless, compared to softmax attention, HDLA still maintains a fundamental linear advantage in terms of linear complexity, and its decode throughput can reach about 89.1% of GLA.

## 5 CONCLUSION

In this work, we propose HDLA, a linear attention mechanism with enhanced structured decay while maintaining reasonable computational and I/O cost, verify its effectiveness across various types of experiments, and obtained its theoretical justification from Test-Time Training perspective. Its robustness demonstrates that more sophisticated, structured and rank-enhanced decay structures can improve the effectiveness of linear attention mechanisms. We've also derived a more general algorithmic framework of linear attention, enabling both diagonal-plus-rank-$r_{ab}$ decay and rank-$r_{kv}$ key-value outer product updates, laying a solid foundation for future research.

**Discussion and Limitation.** Despite its superior experimental performance and sound theoretical explanations, this work has at least the following limitations: (1) In terms of state expansion, this work only explores a naive approach by altering the number of attention heads. Yet, to further bridge the performance gap with Softmax attention, it is necessary to introduce more efficient multi-level and functionally differenciated state expansion methods. (2) Purely linearized hidden state update operations limit the model's expressive power. As suggested in Behrouz et al. (2025c), it is important to appropriately introduce non-linear operations on the hidden state to enhance expressiveness. Nevertheless, HDLA has defined a more efficient utilization mechanism for a single hidden state, and holds significant potential to inspire subsequent research in the rank-enhancement design trends. (3) Due to constrains on computational resource, most pre-training experiments with parameter sizes of 1.45B and 2.8B (except for the 1.45B HDLA and Gated DeltaNet) did not reach the Chinchilla optimal number of tokens (at least 20x the parameter count), resulting in insufficient convergence.

## 6 ACKNOWLEDGEMENT

This work was partially supported by CAS Project for Young Scientists in Basic Research (YSBR-116), National Natural Science Foundation of China (62325603, 62236009, U22A20103), Beijing Science and Technology Plan (Z241100004224011), and shanghai NeuHelium Neuromorphic Technology Co., Ltd.

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

## A    DECLARATION OF LARGE LANGUAGE MODEL (LLM) USAGE

To make the language more fluent and smooth, we've used large language models (LLMs) for polishing during the writing process. We assure that all methods and experiments have been conducted manually and are authentic and valid.

## B    SUPPLEMENTARY EXPERIMENTS

### B.1    STATE EXPANSION EXPERIMENTS

The hidden state size $S$ of linear attention mechanisms can be computed by the following formula, where $n_h$ is the number of attention heads, and $d_k$ and $d_v$ are the total dimensions of the keys and values, respectively:

$$S = n_h \cdot \frac{d_k}{n_h} \cdot \frac{d_v}{n_h} = \frac{d_k d_v}{n_h} \tag{22}$$

Therefore, without changing $d_k$ and $d_v$, we can adjust the hidden state size by altering $n_h$.

**MAD experiment after state expansion.** In table 3, we follow the MAD protocol by setting all linear attention baselines to $n_h = 8$, $d_k = 128$, and $d_v = 128$, resulting in an aligned per-layer hidden state size of $S = 2048$. Here, we naively achieve state expansion by setting $n_h = 4$, so that the per-layer hidden state size of each linear attention model is aligned to $S = 4096$.

Table 10: **Performance comparison on MAD benchmark after expanding the hidden state size from 2048 to 4096.** Mem: Memorization. ICR: In-Context Recall.

| Method | Compression | Fuzzy ICR | ICR | Mem. | Noisy ICR | Selective Copy | AVG. |
|---|---|---|---|---|---|---|---|
| GDP2 | 41.69 | 19.96 | 99.86 | 64.46 | 99.80 | 99.93 | 70.95 |
| DeltaProduct | 42.74 | 21.35 | 99.93 | 52.74 | 99.79 | 99.96 | 69.42 |
| Gated DeltaNet | 44.03 | 18.34 | 99.89 | 66.85 | 99.87 | 95.61 | 70.77 |
| DeltaNet | 43.76 | 24.08 | 99.94 | 42.32 | 99.96 | 99.92 | 68.33 |
| Mamba | 44.82 | 12.21 | 87.24 | 89.25 | 88.74 | 83.08 | 67.56 |
| **HDLA** | 48.47 | 18.34 | 99.99 | 89.24 | 94.42 | 94.55 | **74.17** |

The results in table 10 shows that after state expansion, the average score of HDLA improves by 1.20, still significantly outperforming other linear attention baselines, and is only 1.85% behind Softmax Attention in table 3, demonstrating the effectiveness of HDLA under state expansion.

**Language modeling results of HDLA after state expansion.** The results in table 11 demonstrate that after state expansion, the commonsense reasoning ability of HDLA remains nearly unchanged, while its retrieval performance shows a clear improvement of 2.42% and 3.37%.

| PS (B) | $n_h$ | BQ | PIQA | HS | WG | Arc-e | Arc-c | OBQ | SIQA | SWDE | SC | FDA | AVG-CSR | AVG-RET |
|---|---|---|---|---|---|---|---|---|---|---|---|---|---|---|
| 0.17 | **3** | 58.01 | 63.93 | 33.46 | 50.51 | 53.87 | 25.00 | 30.80 | 37.56 | 9.00 | 17.76 | 2.09 | 44.14 | **9.62** |
| 0.17 | 12 | 59.11 | 64.74 | 33.32 | 49.80 | 52.78 | 26.37 | 30.40 | 36.90 | 4.95 | 15.65 | 1.00 | **44.18** | 7.20 |
| 0.4 | **4** | 61.38 | 66.76 | 39.43 | 49.72 | 59.47 | 29.01 | 34.40 | 39.10 | 13.32 | 23.32 | 5.08 | **47.41** | **13.91** |
| 0.4 | 12 | 58.81 | 67.57 | 39.42 | 51.22 | 60.35 | 28.33 | 33.60 | 38.79 | 8.10 | 21.08 | 2.45 | 47.26 | 10.54 |

Table 11: **Commonsense reasoning and retrieval results of HDLA before and after state expansion.** $d_k = d_v = 768$ at 0.17B parameter scale, while $d_k = d_v = 1024$ at 0.4B parameter scale.

## B.2 HYPERPARAMETER FINE-TUNING EXPERIMENTS

**Fine-tuning experiments on learning rate.** In addition to the 3e-4 learning rate used in the main text, we also compare a range of learning rates (2.0e-4, 2.5e-4, 3.0e-4, and 6.0e-4) following the setup in Dao & Gu (2024). The results show that the models generally perform better with the relatively large learning rate of 6e-4. Due to computational resource constraints, we have not yet applied this setting to the language modeling experiments at 1.45B and 2.8B scales.

| Model | PS(B) | lr | BQ | PIQA | HS | WG | Arc-e | Arc-c | OBQ | SIQA | SWDE | SC | FDA | AVG-CSR | AVG-RET |
|---|---|---|---|---|---|---|---|---|---|---|---|---|---|---|---|
| GDN | 0.2 | 2.0e-4 | 59.4 | 63.1 | 31.9 | 50.0 | 52.0 | 25.3 | 29.6 | 37.2 | 4.8 | 13.2 | 2.0 | 43.6 | 6.7 |
| GDN | 0.2 | 2.5e-4 | 53.8 | 64.0 | 32.3 | 51.0 | 51.9 | 26.0 | 31.2 | 36.4 | 6.1 | 14.3 | 1.4 | 43.3 | 7.3 |
| GDN | 0.2 | 3.0e-4 | 60.1 | 64.1 | 32.7 | 50.0 | 53.6 | 25.3 | 30.8 | 36.7 | 5.9 | 14.3 | 1.0 | 44.2 | 7.1 |
| GDN | 0.2 | 6.0e-4 | 56.9 | 64.4 | 33.9 | 52.8 | 54.6 | 25.1 | 31.2 | 37.1 | 6.6 | 19.7 | 2.5 | 44.5 | 9.6 |
| HDLA | 0.2 | 2.0e-4 | 61.6 | 63.8 | 32.5 | 50.0 | 54.5 | 25.0 | 31.2 | 37.5 | 6.5 | 15.4 | 1.9 | 44.5 | 7.9 |
| HDLA | 0.2 | 2.5e-4 | 61.4 | 64.9 | 33.4 | 49.5 | 55.6 | 26.1 | 30.8 | 37.6 | 6.8 | 17.1 | 1.4 | 44.9 | 8.4 |
| HDLA | 0.2 | 3.0e-4 | 53.8 | 64.3 | 33.5 | 49.9 | 53.6 | 24.9 | 29.8 | 38.6 | 7.7 | 16.9 | 1.1 | 43.5 | 8.5 |
| HDLA | 0.2 | 6.0e-4 | 50.6 | 65.2 | 34.4 | 50.9 | 56.0 | 25.6 | 33.8 | 38.3 | 8.6 | 20.2 | 1.8 | 44.4 | 10.2 |
| GDN | 0.4 | 2.0e-4 | 58.8 | 67.0 | 38.1 | 50.2 | 59.1 | 26.5 | 32.8 | 39.0 | 9.7 | 20.5 | 2.8 | 46.5 | 11.0 |
| GDN | 0.4 | 2.5e-4 | 58.0 | 66.1 | 38.4 | 51.9 | 59.3 | 27.3 | 33.2 | 38.8 | 9.7 | 21.3 | 2.7 | 46.6 | 11.3 |
| GDN | 0.4 | 3.0e-4 | 58.4 | 67.6 | 39.4 | 51.9 | 58.4 | 27.1 | 33.6 | 38.2 | 10.9 | 23.8 | 2.5 | 47.4 | 12.4 |
| GDN | 0.4 | 6.0e-4 | 59.4 | 67.9 | 40.6 | 50.8 | 61.7 | 28.1 | 32.6 | 38.2 | 10.9 | 23.8 | 2.5 | 47.4 | 12.4 |
| HDLA | 0.4 | 2.0e-4 | 60.1 | 66.6 | 39.0 | 50.2 | 59.8 | 28.0 | 33.6 | 37.7 | 9.5 | 22.0 | 2.5 | 46.9 | 11.4 |
| HDLA | 0.4 | 2.5e-4 | 59.4 | 67.5 | 39.3 | 51.1 | 59.7 | 27.7 | 33.4 | 37.8 | 10.0 | 21.8 | 3.7 | 47.0 | 11.8 |
| HDLA | 0.4 | 3.0e-4 | 61.5 | 67.4 | 40.5 | 51.1 | 60.7 | 28.6 | 31.8 | 38.1 | 10.3 | 21.8 | 3.1 | 47.5 | 11.7 |
| HDLA | 0.4 | 6.0e-4 | 57.6 | 67.1 | 41.6 | 50.8 | 60.4 | 27.9 | 33.6 | 39.0 | 12.1 | 23.7 | 4.3 | 47.3 | 13.3 |

Table 12: Commonsense reasoning and retrieval results when fine-tuning on learning rate.

**Fine-tuning experiments on $\beta_t$'s range in HDLA.**

| PS(B) | $\beta_t$ | BQ | PIQA | HS | WG | Arc-e | Arc-c | OBQ | SIQA | SWDE | SC | FDA | AVG-CSR | AVG-RET |
|---|---|---|---|---|---|---|---|---|---|---|---|---|---|---|
| 0.17 | $[0,2]$ | 53.76 | 64.25 | 33.52 | 49.88 | 53.58 | 24.91 | 29.8 | 38.59 | 7.65 | 16.86 | 1.09 | 43.54 | **8.53** |
| 0.17 | $[0,1]$ | 55.11 | 63.66 | 33.33 | 50.83 | 54.84 | 25.77 | 32.2 | 37.46 | 7.29 | 16.99 | 1.09 | **44.15** | 8.46 |
| 0.4 | $[0,2]$ | 61.50 | 67.41 | 40.48 | 51.14 | 60.65 | 28.58 | 31.8 | 38.13 | 10.26 | 21.78 | 3.09 | 47.46 | 11.71 |
| 0.4 | $[0,1]$ | 60.52 | 68.01 | 39.88 | 50.99 | 60.44 | 28.33 | 33.8 | 38.02 | 11.25 | 23.83 | 2.45 | **47.50** | **12.51** |

Table 13: Commonsense reasoning and retrieval results of HDLA under different $\beta_t$ intervals.

**Fine-tuning experiments on the key/value activation function in HDLA.**

| PS(B) | Act | BQ | PIQA | HS | WG | Arc-e | Arc-c | OBQ | SIQA | SWDE | SC | FDA | AVG-CSR | AVG-RET |
|---|---|---|---|---|---|---|---|---|---|---|---|---|---|---|
| 0.17 | SiLU | 53.76 | 64.25 | 33.52 | 49.88 | 53.58 | 24.91 | 29.8 | 38.59 | 7.65 | 16.86 | 1.09 | 43.54 | **8.53** |
| 0.17 | ReLU | 59.30 | 64.36 | 32.91 | 50.51 | 52.74 | 25.26 | 30.8 | 36.80 | 6.39 | 16.62 | 1.00 | 44.09 | 8.00 |
| 0.17 | 1+elu | 59.66 | 64.25 | 32.97 | 50.59 | 53.54 | 24.66 | 30.4 | 37.15 | 6.30 | 17.33 | 1.54 | **44.15** | 8.39 |
| 0.4 | SiLU | 61.50 | 67.41 | 40.48 | 51.14 | 60.65 | 28.58 | 31.8 | 38.13 | 10.26 | 21.78 | 3.09 | **47.46** | 11.71 |
| 0.4 | ReLU | 61.71 | 67.63 | 39.65 | 51.78 | 59.68 | 27.56 | 31.0 | 38.49 | 10.53 | 20.44 | 2.54 | 47.19 | 11.17 |
| 0.4 | 1+elu | 61.28 | 65.94 | 39.00 | 50.67 | 59.51 | 27.82 | 35.8 | 37.72 | 10.62 | 24.80 | 3.36 | 47.22 | **12.93** |

Table 14: Commonsense reasoning and retrieval results of HDLA with different activation functions.

# C  DETAILED ALGORITHMIC RESULTS

## C.1  THE EXACT ARCHITECTURE OF HDLA AS A TOKEN MIXER

Here we present some details omitted in fig. 1 using mathematical formulae:

$$\mathbf{q}_t = \boldsymbol{\theta}_Q \mathbf{x}_t \in \mathbb{R}^{d_k \times 1}, \mathbf{k}_t = \boldsymbol{\theta}_K \mathbf{x}_t \in \mathbb{R}^{d_k \times 1}, \mathbf{v}_t = \boldsymbol{\theta}_V \mathbf{x}_t \in \mathbb{R}^{d_v \times 1} \tag{23}$$

$$\beta_t = \boldsymbol{\theta}_\beta \mathbf{x}_t \in \mathbb{R}, \lambda_t = \boldsymbol{\theta}_\lambda \mathbf{x}_t \in \mathbb{R}^{d_k \times 1} \tag{24}$$

$$\mathbf{q}_t = \mathrm{SiLU}(\mathbf{q}_t), \mathbf{k}_t = \mathrm{SiLU}(\mathbf{k}_t), \mathbf{v}_t = \mathrm{SiLU}(\mathbf{v}_t) \tag{25}$$

$$\mathbf{k}_t = \mathrm{norm}_{l_2}(\mathbf{k}_t) \tag{26}$$

$$\mathbf{S}_t = (\mathbf{I} - \beta_t \mathbf{k}_t \mathbf{k}_t^\top) \mathbf{Diag}(\lambda_t)(\mathbf{I} - \beta_t \mathbf{k}_t \mathbf{k}_t^\top) \mathbf{S}_{t-1} + \mathbf{k}_t \mathbf{v}_t^\top \tag{27}$$

$$\mathbf{y}_t = \mathbf{q}_t \mathbf{S}_t \in \mathbb{R}^{d_v \times 1} \tag{28}$$

$$\mathbf{g}_t = \mathbf{x}_t \boldsymbol{\theta}_g \in \mathbb{R}^{d_v \times 1} \tag{29}$$

$$\mathbf{o}_t = \mathbf{y}_t \odot \mathbf{g}_t \in \mathbb{R}^{d_v \times 1} \tag{30}$$

## C.2  COMPUTATION OF $\mathbf{A}_t$ AND $\mathbf{B}_t$

Consider factorizing each element of $\boldsymbol{\Lambda}_t$ as the product of its square roots, then coupling them with the left and right Householder transformations $\mathbf{H}_t$. This yields the reformulation of $\mathbf{P}_t$ as follows:

$$\mathbf{P}_t = \left( \mathrm{Diag}(\sqrt{\lambda_t}) - \beta_t \mathbf{k}_t (\mathbf{k}_t \odot \sqrt{\lambda_t})^\top \right) \left( \mathrm{Diag}(\sqrt{\lambda_t}) - \beta_t (\mathbf{k}_t \odot \sqrt{\lambda_t}) \mathbf{k}_t^\top \right) \tag{31}$$

This form of $\mathbf{P}_t$ is a special case of two diagonal-plus-rank-one matrices' product:

$$\mathbf{P}_t = (\mathbf{D}_{t,(1)} - \mathbf{a}_{t,(1)} \mathbf{b}_{t,(1)}^\top)(\mathbf{D}_{t,(2)} - \mathbf{a}_{t,(2)} \mathbf{b}_{t,(2)}^\top) \tag{32}$$

Utilizing the compact WY representation of diagonal-plus-rank-1 matrices' cumulative products (Yang & Zhang, 2024), $\mathbf{P}_t$ can be rewritten as the following form:

$$\mathbf{P}_t = \mathbf{D}_t - \mathbf{A}_t \mathbf{B}_t^\top \in \mathbb{R}^{d_k \times d_k}, \mathbf{A}_t, \mathbf{B}_t \in \mathbb{R}^{d_k \times 2} \tag{33}$$

We demonstrate the detailed formulation of $\mathbf{A}_t, \mathbf{B}_t$ as follows:

$$\mathbf{P}_t = (\mathbf{D}_{t,(1)} - \mathbf{a}_{t,(1)}\mathbf{b}_{t,(1)}^\top)(\mathbf{D}_{t,(2)} - \mathbf{a}_{t,(2)}\mathbf{b}_{t,(2)}^\top)$$
$$= \mathbf{D}_{t,(1)}\mathbf{D}_{t,(2)} - \mathbf{a}_{t,(1)}\mathbf{b}_{t,(1)}^\top\mathbf{D}_{t,(2)} - (\mathbf{D}_{t,(1)} - \mathbf{a}_{t,(1)}\mathbf{b}_{t,(1)}^\top)\mathbf{a}_{t,(2)}\mathbf{b}_{t,(2)}^\top$$

Let:

$$\mathbf{D}_t = \mathbf{D}_{t,(1)}\mathbf{D}_{t,(2)} \in \mathbb{R}^{d_k \times d_k} \tag{34}$$

$$\mathbf{A}_t = \begin{bmatrix} \mathbf{a}_{t,(1)} & (\mathbf{D}_{t,(1)} - \mathbf{a}_{t,(1)}\mathbf{b}_{t,(1)}^\top)\mathbf{a}_{t,(2)} \end{bmatrix} \in \mathbb{R}^{d_k \times 2} \tag{35}$$

$$\mathbf{B}_t = \begin{bmatrix} \mathbf{D}_{t,(2)}\mathbf{b}_{t,(1)} & \mathbf{b}_{t,(2)} \end{bmatrix} \in \mathbb{R}^{d_k \times 2} \tag{36}$$

Then $\mathbf{P}_t$ can be rewritten as eq. (33)'s form.

## C.3 SUPPLEMENTARY DEDUCTION OF THE FORWARD CHUNK-WISE PARALLEL ALGORITHM

For deduction of cumulative products of the decay matrices $\mathbf{P}_1, \mathbf{P}_2, \cdots$, observe that:

$$\mathbf{P}_1 = \mathbf{D}_1 - \mathbf{A}_1\mathbf{B}_1^\top$$

$$\mathbf{P}_2 = (\mathbf{D}_1 - \mathbf{A}_1\mathbf{B}_1^\top)(\mathbf{D}_2 - \mathbf{A}_2\mathbf{B}_2^\top)$$
$$= \mathbf{D}_1\mathbf{D}_2 - \mathbf{A}_1\mathbf{B}_1^\top\mathbf{D}_2 - (\mathbf{D}_1 - \mathbf{A}_1\mathbf{B}_1)\mathbf{A}_2\mathbf{B}_2^\top$$

$$\mathbf{P}_3 = (\mathbf{D}_1\mathbf{D}_2 - \mathbf{A}_1\mathbf{B}_1^\top\mathbf{D}_2 - (\mathbf{D}_1 - \mathbf{A}_1\mathbf{B}_1)\mathbf{A}_2\mathbf{B}_2^\top)(\mathbf{D}_3 - \mathbf{A}_3\mathbf{B}_3^\top)$$
$$= \mathbf{D}_1\mathbf{D}_2\mathbf{D}_3 - \mathbf{A}_1\mathbf{B}_1^\top\mathbf{D}_2\mathbf{D}_3$$
$$- (\mathbf{D}_1 - \mathbf{A}_1\mathbf{B}_1)\mathbf{A}_2\mathbf{B}_2^\top\mathbf{D}_3 - \big(\mathbf{D}_1\mathbf{D}_2 - \mathbf{A}_1\mathbf{B}_1^\top\mathbf{D}_2 - (\mathbf{D}_1 - \mathbf{A}_1\mathbf{B}_1)\mathbf{A}_2\mathbf{B}_2^\top\big)\mathbf{A}_3\mathbf{B}_3^\top$$

Suppose:

$$\mathbf{P}_t = \mathbf{D}_0^t - \sum_{i=1}^{t} \mathbf{W}_i\mathbf{B}_i^\top\mathbf{D}_i^t$$

Then:

$$\mathbf{P}_{t+1} = (\mathbf{D}_0^t - \sum_{i=1}^{t} \mathbf{W}_i\mathbf{B}_i^\top\mathbf{D}_i^t)(\mathbf{D}_{t+1} - \mathbf{A}_{t+1}\mathbf{B}_{t+1}^\top)$$
$$= \mathbf{D}_0^{t+1} - \sum_{i=1}^{t} \mathbf{W}_i\mathbf{B}_i^\top\mathbf{D}_i^{t+1} - \left( (\mathbf{D}_0^t - \sum_{i=1}^{t} \mathbf{W}_i\mathbf{B}_i^\top\mathbf{D}_i^t)\mathbf{A}_{t+1} \right)\mathbf{B}_{t+1}^\top\mathbf{D}_{t+1}^{t+1}$$
$$= \mathbf{D}_0^{t+1} - \sum_{i=1}^{t+1} \mathbf{W}_i\mathbf{B}_i^\top\mathbf{D}_i^{t+1}$$
$$= (\mathbf{I} - \sum_{i=1}^{t+1} \mathbf{W}_i\mathbf{B}_i^{'\top})\mathbf{D}_0^{t+1}$$

Where:

$$\mathbf{W}_t = (\mathbf{D}_0^{t-1} - \sum_{i=1}^{t-1} \mathbf{W}_i \mathbf{B}_i^\top \mathbf{D}_i^{t-1}) \mathbf{A}_t$$

$$= (\mathbf{I} - \sum_{i=1}^{t-1} \mathbf{W}_i (\frac{\mathbf{B}_i}{\mathbf{D}_0^i})^\top)(\mathbf{A}_t \odot \mathbf{D}_0^{t-1})$$

$$= (\mathbf{I} - \sum_{i=1}^{t-1} \mathbf{W}_i \mathbf{B}_i'^\top) \mathbf{A}_t'$$

For compact form of $\mathbf{S}_1, \mathbf{S}_2, \cdots$, observe that:

$$\mathbf{S}_1 = \mathbf{V}_1 \mathbf{K}_1^\top$$

$$\mathbf{S}_2 = \mathbf{S}_1 (\mathbf{D}_1^2 - \mathbf{A}_2 \mathbf{B}_2^\top) + \mathbf{V}_2 \mathbf{K}_2^\top$$
$$= \mathbf{V}_1 \mathbf{K}_1^\top \mathbf{D}_1^2 - \mathbf{V}_1 \mathbf{K}_1^\top \mathbf{A}_2 \mathbf{B}_2^\top + \mathbf{V}_2 \mathbf{K}_2^\top$$

$$\mathbf{S}_3 = \mathbf{S}_2 (\mathbf{D}_2^3 - \mathbf{A}_3 \mathbf{B}_3^\top) + \mathbf{V}_3 \mathbf{K}_3^\top$$
$$= \mathbf{V}_1 \mathbf{K}_1^\top \mathbf{D}_1^3 + \mathbf{V}_2 \mathbf{K}_2^\top \mathbf{D}_2^3 + \mathbf{V}_3 \mathbf{K}_3^\top - \mathbf{V}_1 \mathbf{K}_1^\top \mathbf{A}_2 \mathbf{B}_2^\top \mathbf{D}_2^3$$
$$- \left( (\mathbf{V}_1 \mathbf{K}_1^\top \mathbf{D}_1^2 - \mathbf{V}_1 \mathbf{K}_1^\top \mathbf{A}_2 \mathbf{B}_2^\top + \mathbf{V}_2 \mathbf{K}_2^\top) \mathbf{A}_3 \right) \mathbf{B}_3^\top$$

Suppose:

$$\mathbf{S}_t = \sum_{i=1}^t (\mathbf{V}_i \mathbf{K}_i^\top - \mathbf{U}_i \mathbf{B}_i^\top) \mathbf{D}_i^t$$

Then:

$$\mathbf{S}_{t+1} = \mathbf{S}_t (\mathbf{D}_t^{t+1} - \mathbf{A}_{t+1} \mathbf{B}_{t+1}^\top) + \mathbf{V}_{t+1} \mathbf{K}_{t+1}^\top$$
$$= \sum_{i=1}^t (\mathbf{V}_i \mathbf{K}_i^\top - \mathbf{U}_i \mathbf{B}_i^\top) \mathbf{D}_i^{t+1} + \mathbf{V}_{t+1} \mathbf{K}_{t+1}^\top - \left( \sum_{i=1}^t (\mathbf{V}_i \mathbf{K}_i^\top - \mathbf{U}_i \mathbf{B}_i^\top) \mathbf{D}_i^t \mathbf{A}_{t+1} \right) \mathbf{B}_{t+1}^\top$$
$$= \sum_{i=1}^{t+1} (\mathbf{V}_i \mathbf{K}_i^\top - \mathbf{U}_i \mathbf{B}_i^\top) \mathbf{D}_i^{t+1}$$
$$= \sum_{i=1}^{t+1} (\mathbf{V}_i \mathbf{K}_i'^\top - \mathbf{U}_i \mathbf{B}_i'^\top) \mathbf{D}_0^{t+1}$$

Where:

$$\mathbf{U}_t = \sum_{i=1}^{t-1} (\mathbf{V}_i \mathbf{K}_i^\top - \mathbf{U}_i \mathbf{B}_i^\top) \mathbf{D}_i^{t-1} \mathbf{A}_t$$
$$= \sum_{i=1}^{t-1} (\mathbf{V}_i \mathbf{K}_i'^\top - \mathbf{U}_i \mathbf{B}_i'^\top) \mathbf{A}_t'$$

## C.4 BACKWARD CHUNK-WISE PARALLEL ALGORITHM FOR LINEAR ATTENTION WITH DIAGONAL-PLUS-RANK-$r_{ab}$ DECAY STRUCTURE AND RANK-$r_{kv}$ KEY-VALUE UPDATES

For the sake of simplicity, make the following definitions:

$$\mathbf{\Lambda}_{[n]} = \begin{bmatrix} \mathbf{d}_{(n-1)C}^{(n-1)C+1} & \mathbf{d}_{(n-1)C}^{(n-1)C+2} & \cdots & \mathbf{d}_{(n-1)C}^{nC} \end{bmatrix} \in \mathbb{R}^{d_k \times C} \tag{37}$$

$$\bar{\mathbf{\Lambda}}_{[n]} = \begin{bmatrix} \mathbf{d}_{(n-1)C+1}^{nC} & \mathbf{d}_{(n-1)C+2}^{nC} & \cdots & \mathbf{d}_{nC}^{nC} \end{bmatrix} \in \mathbb{R}^{d_k \times C} \tag{38}$$

$$\tilde{\mathbf{\Lambda}}_{[n]} = \begin{bmatrix} \mathbf{d}_{(n-1)C}^{(n-1)C} & \mathbf{d}_{(n-1)C}^{(n-1)C+1} & \cdots & \mathbf{d}_{(n-1)C}^{nC-1} \end{bmatrix} \in \mathbb{R}^{d_k \times C} \tag{39}$$

For $\bar{\mathbf{\Lambda}}_{[n]}, \tilde{\mathbf{\Lambda}}_{[n]}$'s $\odot$ operation with $\mathbf{K}_{[n]}, \mathbf{A}_{[n]}$ or $\mathbf{B}_{[n]}, \bar{\mathbf{\Lambda}}_{[n]}, \tilde{\mathbf{\Lambda}}_{[n]}$ are repeated across the sequential dimension in an interleaving manner (analogous to torch.repeat_interleave) before taking element-wise multiplications.

Let's review the forward formulae of linear attention with diagonal-plus-rank-$r_{ab}$ decay structure and rank-$r_{kv}$ key-value updates:

$$\mathbf{U}_{[n]} = \mathbf{V}_{[n]}\text{triu}_{r_{kv} \times r_{ab}}(\mathbf{K}_{[n]}'\mathbf{A}_{[n]}'^{\top}, 1)\left(\mathbf{I} + \text{triu}_{r_{ab} \times r_{ab}}(\mathbf{B}_{[n]}'\mathbf{A}_{[n]}'^{\top}, 1)\right)^{-1} \in \mathbb{R}^{d_v \times r_{ab}C}, \tag{19}$$

$$\mathbf{W}_{[n]} = \mathbf{A}_{[n]}'\left(\mathbf{I} + \text{triu}_{r_{ab} \times r_{ab}}(\mathbf{B}_{[n]}'\mathbf{A}_{[n]}'^{\top}, 1)\right)^{-1} \in \mathbb{R}^{d_k \times r_{ab}C}, \tag{20}$$

$$\mathbf{C}_{[n]} = \mathbf{S}_{[n-1]}^{\top}\mathbf{W}_{[n]} + \mathbf{U}_{[n]} \tag{40}$$

$$\mathbf{O}_{[n]} = \mathbf{S}_{[n-1]}^{\top}\mathbf{Q}_{[n]}' + \mathbf{V}_{[n]}\text{triu}_{r_{kv} \times 1}(\mathbf{K}_{[n]}'^{\top}\mathbf{Q}_{[n]}', 0) - \mathbf{C}_{[n]}\text{triu}_{r_{ab} \times 1}(\mathbf{B}_{[n]}'\mathbf{Q}_{[n]}', 0), \tag{41}$$

$$\mathbf{S}_{[n]} = \mathbf{D}_{(n-1)C}^{nC}\mathbf{S}_{[n-1]} + (\mathbf{K}_{[n]} \odot \bar{\mathbf{\Lambda}}_{[n]})\mathbf{V}_{[n]}^{\top} - (\mathbf{B}_{[n]} \odot \bar{\mathbf{\Lambda}}_{[n]})\mathbf{C}_{[n]}^{\top}, \tag{42}$$

In the following subsections, we deduct the gradient of $\mathbf{C}_{[n]}$ and $\mathbf{S}_{[n-1]}$ first, which needs to be computed serially. Then, the gradient of $\mathbf{Q}_{[n]}, \mathbf{K}_{[n]}, \mathbf{V}_{[n]}, \mathbf{A}_{[n]}, \mathbf{B}_{[n]}$ of each chunk can be computed in parallel. Finally, we will derive a concise form of the decay matrices' diagonal term.

### C.4.1 Deduction of $\partial\mathbf{C}_{[n]}$ and $\partial\mathbf{S}_{[n-1]}$

Since $\mathbf{C}_{[n]}$ participates in the computation of $\mathbf{O}_{[n]}$ (eq. (41)) and the update of $\mathbf{S}_{[n]}$ (eq. (42)), its gradient is composed of two parts:

$$\partial\mathbf{C}_{[n]} = \underbrace{-\partial\mathbf{O}_{[n]}\text{tril}_{1 \times r_{ab}}(\mathbf{Q}_{[n]}'^{\top}\mathbf{B}_{[n]}', 0)}_{\partial\mathbf{C}_{[n],\text{intra}}} + \underbrace{\partial\mathbf{S}_{[n]}^{\top}(\mathbf{B}_{[n]} \odot \bar{\mathbf{\Lambda}}_{[n]})}_{\partial\mathbf{C}_{[n],\text{inter}}} \tag{43}$$

Since $\mathbf{S}_{[n-1]}$ participates in the computation of $\mathbf{C}_{[n]}, \mathbf{O}_{[n]}$, as well as the update of $\mathbf{S}_{[n]}$, the gradient of $\mathbf{S}_{[n-1]}$ is:

$$\partial\mathbf{S}_{[n-1]} = \underbrace{\mathbf{W}_{[n]}\partial\mathbf{C}_{[n]}^{\top} + \mathbf{Q}_{[n]}\partial\mathbf{O}_{[n]}^{\top}}_{\partial\mathbf{S}_{[n-1],\text{intra}}} + \underbrace{\mathbf{D}_{(n-1)C}^{nC}\partial\mathbf{S}_{[n]}}_{\partial\mathbf{S}_{[n-1],\text{inter}}} \tag{44}$$

### C.4.2 Deduction of $\partial\mathbf{Q}_{[n]}, \partial\mathbf{K}_{[n]}, \partial\mathbf{V}_{[n]}, \partial\mathbf{A}_{[n]}$ and $\partial\mathbf{B}_{[n]}$

Since $\mathbf{Q}_{[n]}$ participates in the computation of $\mathbf{O}_{[n]}$, both intra-chunk and inter-chunk, its gradidient is composed of:

$$\partial\mathbf{Q}_{[n],\text{intra,part1}} = (\mathbf{K}_{[n]} \odot \bar{\mathbf{\Lambda}}_{[n]})\text{triu}_{r_{kv} \times 1}(\mathbf{V}_{[n]}^{\top}\partial\mathbf{O}_{[n]}, 0) \tag{45}$$

$$\partial\mathbf{Q}_{[n],\text{intra,part2}} = -(\mathbf{B}_{[n]} \odot \bar{\mathbf{\Lambda}}_{[n]})\text{triu}_{r_{ab} \times 1}(\mathbf{C}_{[n]}^{\top}\partial\mathbf{O}_{[n]}, 0) \tag{46}$$

$$\partial\mathbf{Q}_{[n],\text{inter}} = (\mathbf{S}_{[n-1]}\partial\mathbf{O}_{[n]}) \odot \mathbf{\Lambda}_{[n]} \tag{47}$$

$$\partial\mathbf{Q}_{[n]} = \partial\mathbf{Q}_{[n],\text{intra,part1}} + \partial\mathbf{Q}_{[n],\text{intra,part2}} + \partial\mathbf{Q}_{[n],\text{inter}} \tag{48}$$

Similarly, the gradients of $\mathbf{K}_{[n]}, \mathbf{B}_{[n]}, \mathbf{V}_{[n]}$ relevant to rank-enhanced gated linear attention can be derived as follows:

$$\partial\mathbf{K}_{[n],\text{gla}} = \underbrace{\left((\mathbf{Q}_{[n]} \odot \mathbf{\Lambda}_{[n]})\text{tril}_{1 \times r_{kv}}(\partial\mathbf{O}_{[n]}^{\top}\mathbf{V}_{[n]}, 0)\right) \oslash \mathbf{\Lambda}_{[n]}}_{\partial\mathbf{K}_{[n],\text{gla,intra}}} + \underbrace{(\partial\mathbf{S}_{[n]}\mathbf{V}_{[n]}) \odot \bar{\mathbf{\Lambda}}_{[n]}}_{\partial\mathbf{K}_{[n],\text{gla,inter}}} \tag{49}$$

$$\partial\mathbf{B}_{[n],\text{gla}} = \underbrace{-\left((\mathbf{Q}_{[n]} \odot \mathbf{\Lambda}_{[n]})\text{tril}_{1 \times r_{kv}}(\partial\mathbf{O}_{[n]}^{\top}\mathbf{C}_{[n]}, 0)\right) \oslash \mathbf{\Lambda}_{[n]}}_{\partial\mathbf{B}_{[n],\text{gla,intra}}} + \underbrace{\left(-(\partial\mathbf{S}_{[n]}\mathbf{C}_{[n]}) \odot \bar{\mathbf{\Lambda}}_{[n]}\right)}_{\partial\mathbf{B}_{[n],\text{gla,inter}}} \tag{50}$$

$$\partial\mathbf{V}_{[n],\text{gla}} = \underbrace{\partial\mathbf{O}_{[n]}\text{tril}_{1 \times r_{kv}}(\mathbf{Q}_{[n]}^{'\top}\mathbf{K}_{[n]}^{'}, 0)}_{\partial\mathbf{V}_{[n],\text{intra}}} + \underbrace{\mathbf{S}_{[n]}^{\top}\mathbf{D}_{(n-1)C}^{nC}\mathbf{K}_{[n]}^{'}}_{\partial\mathbf{V}_{[n],\text{intra}}} \tag{51}$$

$$\partial\mathbf{V}_{[n],\text{gla}} = \underbrace{-\partial\mathbf{O}_{[n]}\text{tril}_{1 \times r_{ab}}(\mathbf{Q}_{[n]}^{'\top}\mathbf{B}_{[n]}^{'}, 0)}_{\partial\mathbf{C}_{[n],\text{intra}}} + \underbrace{\mathbf{S}_{[n]}^{\top}\mathbf{D}_{(n-1)C}^{nC}\mathbf{B}_{[n]}^{'}}_{\partial\mathbf{C}_{[n],\text{intra}}} \tag{52}$$

Now we consider the gradients of each input corresponding to the arbitrary-rank WY compact representation (eq. (13) and eq. (14)). The matrix inversion operation can be avoided in the backward pass, utilizing the following observation of $\mathbf{u}_t$'s recurrent definition.

$$\mathbf{u}_t = \sum_{i=1}^{t-1}(\mathbf{v}_i\mathbf{k}_i^{'\top} - \mathbf{u}_i\mathbf{b}_i^{'\top})\mathbf{a}_t^{'}$$

$$\Rightarrow \partial\mathbf{a}_{t,\text{part1}}^{'} = \sum_{i=1}^{t-1}(\mathbf{k}_i^{'}\mathbf{v}_i^{\top} - \mathbf{b}_i^{'}\mathbf{u}_i^{\top})\partial\mathbf{u}_t$$

$$\Rightarrow \partial\mathbf{A}_{\text{part1}}^{'} = \mathbf{K}_{[n]}^{'}\text{triu}_{r_{kv} \times r_{ab}}(\mathbf{V}_{[n]}^{\top}\partial\mathbf{U}_{[n]}, 1) - \mathbf{B}_{[n]}^{'}\text{triu}_{r_{ab} \times r_{ab}}(\mathbf{U}_{[n]}^{\top}\partial\mathbf{U}_{[n]}, 1)$$

$$\Rightarrow \partial\mathbf{A}_{[n],\text{part1}} = \mathbf{\Lambda}_{[n]} \odot \left((\mathbf{K}_{[n]} \oslash \mathbf{\Lambda}_{[n]})\text{triu}_{r_{kv} \times r_{ab}}(\mathbf{V}_{[n]}^{\top}\partial\mathbf{U}_{[n]}, 1)\right)$$
$$- \mathbf{\Lambda}_{[n]} \odot \left((\mathbf{B}_{[n]} \oslash \mathbf{\Lambda}_{[n]})\text{triu}_{r_{ab} \times r_{ab}}(\mathbf{U}_{[n]}^{\top}\partial\mathbf{U}_{[n]}, 1)\right) \tag{53}$$

$$\mathbf{w}_t = (\mathbf{I} - \sum_{i=1}^{t-1}\mathbf{w}_i(\mathbf{b}_i \oslash \mathbf{D}_0^i)^{\top})(\mathbf{a}_t \odot \mathbf{D}_0^{t-1})$$

$$\Rightarrow \partial(\mathbf{a}_t \odot \mathbf{D}_0^{t-1}) = (\mathbf{I} - \sum_{i=1}^{t-1}(\mathbf{b}_i \oslash \mathbf{D}_0^i)\mathbf{w}_i^{\top})\partial\mathbf{w}_t$$

$$\Rightarrow \partial(\mathbf{A}_{[n]} \odot \tilde{\mathbf{\Lambda}}_{[n]}) = \partial\mathbf{W}_{[n]} - \mathbf{B}_{[n]} \oslash \mathbf{\Lambda}_{[n]}\text{triu}(\mathbf{W}_{[n]}^{\top}\partial\mathbf{W}_{[n]}, 1)$$

$$\Rightarrow \partial\mathbf{A}_{[n],\text{part2}} = \tilde{\mathbf{\Lambda}}_{[n]} \odot \left(\partial\mathbf{W}_{[n]} - (\mathbf{B}_{[n]} \oslash \mathbf{\Lambda}_{[n]})\text{triu}_{r_{ab} \times r_{ab}}(\mathbf{W}_{[n]}^{\top}\partial\mathbf{W}_{[n]}, 1)\right) \tag{54}$$

$$\partial\mathbf{A}_{[n]} = \partial\mathbf{A}_{[n],\text{part1}} + \partial\mathbf{A}_{[n],\text{part2}} \tag{55}$$

$$\mathbf{u}_t = \sum_{i=1}^{t-1}(\mathbf{v}_i(\mathbf{k}_i \oslash \mathbf{D}_0^i)^{\top} - \mathbf{u}_i(\mathbf{b}_i \oslash \mathbf{D}_0^i)^{\top})(\mathbf{a}_t \odot \mathbf{D}_0^{t-1})$$

$$\Rightarrow \partial(\mathbf{b}_i \oslash \mathbf{D}_0^i)^{\top} = -\mathbf{u}_i^{\top}\mathbf{du}_t(\mathbf{a}_t \odot \mathbf{D}_0^{t-1})^{\top}$$

$$\Rightarrow \partial\mathbf{b}_{i,\text{wy,part1}} = -(\mathbf{1} \oslash D_0^i)\sum_{j=i+1}^{t}\left((\mathbf{a}_j \odot D_0^{j-1})\mathbf{du}_j^{\top}\mathbf{u}_i\right)$$

$$\Rightarrow \partial\mathbf{B}_{[n],\text{wy,part1}} = -(\mathbf{1} \oslash \mathbf{\Lambda}_{[n]})\left((\mathbf{A}_{[n]} \odot \tilde{\mathbf{\Lambda}}_{[n]})\text{tril}_{r_{ab} \times r_{ab}}(\partial\mathbf{U}_{[n]}^{\top}\mathbf{U}_{[n]}, -1)\right) \tag{56}$$

$$\mathbf{w}_t = \left(\mathbf{I} - \sum_{i=1}^{t-1} \mathbf{w}_i(\mathbf{b}_i \oslash \mathbf{D}_0^i)^\top\right)(\mathbf{a}_t \odot \mathbf{D}_0^{t-1})$$

$$\Rightarrow \partial(\mathbf{b}_i \oslash \mathbf{D}_0^i)^\top = -\mathbf{w}_i^\top \partial\mathbf{w}_t(\mathbf{a}_t \odot \mathbf{D}_0^{t-1})^\top$$

$$\Rightarrow \partial\mathbf{b}_{i,\mathrm{wy},\mathrm{part2}} = -(\mathbf{1} \oslash D_0^i) \sum_{j=i+1}^{t} \left((\mathbf{a}_j \odot D_0^{j-1})\partial\mathbf{w}_j^\top \mathbf{w}_i\right)$$

$$\Rightarrow \partial\mathbf{B}_{[n],\mathrm{wy},\mathrm{part2}} = -(\mathbf{1} \oslash \mathbf{\Lambda}_{[n]})\left((\mathbf{A}_{[n]} \odot \tilde{\mathbf{\Lambda}}_{[n]})\mathrm{tril}_{r_{ab} \times r_{ab}}(\partial\mathbf{W}_{[n]}^\top \mathbf{W}_{[n]}, -1)\right) \tag{57}$$

$$\partial\mathbf{B}_{[n],\mathrm{wy}} = -(1 \oslash \mathbf{\Lambda}_{[n]})\left((\mathbf{A}_{[n]} \odot \tilde{\mathbf{\Lambda}}_{[n]})\mathrm{tril}_{r_{ab} \times r_{ab}}(\partial\mathbf{U}_{[n]}^\top \mathbf{U}_{[n]}, -1)\right)$$
$$- (1 \oslash \mathbf{\Lambda}_{[n]})\left((\mathbf{A}_{[n]} \odot \tilde{\mathbf{\Lambda}}_{[n]})\mathrm{tril}_{r_{ab} \times r_{ab}}(\partial\mathbf{W}_{[n]}^\top \mathbf{W}_{[n]}, -1)\right) \tag{58}$$

Similar observation and deduction yields the following results:

$$\partial\mathbf{k}_{i,\mathrm{wy}} = \sum_{j=i+1}^{t} \mathbf{D}_i^{j-1}\mathbf{a}_j\partial\mathbf{u}_j^\top \mathbf{v}_i$$

$$= (\mathbf{1} \oslash \mathbf{D}_0^i) \sum_{j=i+1}^{t} (\mathbf{D}_0^{j-1}\mathbf{a}_j)\partial\mathbf{u}_j^\top \mathbf{v}_i$$

$$= (\mathbf{1} \oslash \mathbf{D}_0^i) \sum_{j=i+1}^{t} \left((\mathbf{a}_j')\partial\mathbf{u}_j^\top\right)\mathbf{v}_i$$

$$\Rightarrow \partial\mathbf{K}_{[n],\mathrm{wy}} = \begin{bmatrix} 1 \oslash \mathbf{D}_0^1 & 1 \oslash \mathbf{D}_0^2 & \cdots & 1 \oslash \mathbf{D}_0^t \end{bmatrix} \odot \left(\mathbf{A}_{[n]}'\mathrm{tril}(\partial\mathbf{U}_{[n]}^\top \mathbf{V}_{[n]}, -1)\right)$$

$$\Rightarrow \partial\mathbf{K}_{[n],\mathrm{wy}} = (1 \oslash \mathbf{\Lambda}_{[n]}) \odot \left((\mathbf{A}_{[n]} \odot \tilde{\mathbf{\Lambda}}_{[n]})\mathrm{tril}_{r_{ab} \times r_{kv}}(\partial\mathbf{U}_{[n]}^\top \mathbf{V}_{[n]}, -1)\right) \tag{59}$$

$$\mathbf{u}_t = \sum_{i=1}^{t-1}(\mathbf{v}_i(\mathbf{k}_i \oslash \mathbf{D}_0^i)^\top - \mathbf{u}_i(\mathbf{b}_i \oslash \mathbf{D}_0^i)^\top)(\mathbf{a}_t \odot \mathbf{D}_0^{t-1})$$

$$\Rightarrow \partial\mathbf{v}_{i,\mathrm{wy}} = \sum_{j=i+1}^{t} \partial\mathbf{u}_j(\mathbf{a}_j \odot D_0^{j-1})^\top(\mathbf{k}_i \oslash D_0^i)$$

$$\Rightarrow \partial\mathbf{V}_{[n],\mathrm{wy}} = \partial\mathbf{U}_{[n]}\mathrm{tril}_{r_{ab} \times r_{kv}}\left((\mathbf{A}_{[n]} \odot \tilde{\mathbf{\Lambda}}_{[n]})^\top(\mathbf{K}_{[n]} \oslash \mathbf{\Lambda}_{[n]}), -1\right) \tag{60}$$

If a tensor participates in the computation of WY representation and rank-enhanced gated linear attention at the same time, then its total gradient is the summation of gradients relevant to the former and the latter computations. For example:

$$\partial\mathbf{V}_{[n]} = \partial\mathbf{V}_{[n],\mathrm{gla}} + \partial\mathbf{V}_{[n],\mathrm{wy}} \tag{61}$$

### C.4.3 DEDUCTION OF $\partial\mathbf{\Lambda}_{[n]}$ AND $\partial\tilde{\mathbf{\Lambda}}_{[n]}$

Notice that the diagonal terms of Diagonal-Plus-Rank-$r_{ab}$ decay matrices are applied on other input tensors, using their chunk-wise cumulative products, and they are applied on $\mathbf{Q}_{[n]}$ and $\mathbf{A}_{[n]}$ ($\mathbf{K}_{[n]}$ and $\mathbf{B}_{[n]}$) in element-wise multiplication (division) manner:

$$\mathbf{Q}^{'}_{[n],:,t} = \mathbf{Q}_{[n],:,t} \odot \mathbf{d}^{(n-1)C+t}_{(n-1)C}$$

$$\mathbf{A}^{'}_{[n],:,t \cdot r_{ab}+r} = \mathbf{A}_{[n],:,t \cdot r_{ab}+r} \odot \mathbf{d}^{(n-1)C+(t-1)}_{(n-1)C}$$

$$\mathbf{B}^{'}_{[n],:,t \cdot r_{ab}+r} = \mathbf{B}_{[n],:,t \cdot r_{ab}+r} \oslash \mathbf{d}^{(n-1)C+t}_{(n-1)C}$$

$$\mathbf{K}^{'}_{[n],:,t \cdot r_{kv}+r} = \mathbf{K}_{[n],:,t \cdot r_{kv}+r} \oslash \mathbf{d}^{(n-1)C+t}_{(n-1)C}$$

Now, we'd like to derive the gradient for the chunk-wise cumulative sum of the logarithms of diagonal decay terms, i.e., $\mathbf{d}^{(n-1)C+1}_{(n-1)C}, \mathbf{d}^{(n-1)C+2}_{(n-1)C+1}, \cdots$.

For simplicity, consider the special case when $r_{ab} = 1$, and the result can be easily generalized to arbitrarily chosen $r_{ab}$.

Let's derive the gradient of $\mathbf{d}^{(n-1)C+t}_{(n-1)C}$ and $\mathbf{d}^{(n-1)C+t-1}_{(n-1)C}$ first, for each $1 \le t \le C$:

$$\partial\mathbf{d}^{(n-1)C+t}_{(n-1)C} = (\partial\mathbf{q}^{'}_{(n-1)C+t} \odot \mathbf{q}_{(n-1)C+t})$$
$$- (\partial\mathbf{k}^{'}_{(n-1)C+t} \odot \mathbf{k}_{(n-1)C+t}) \oslash (\mathbf{d}^{(n-1)C+t}_{(n-1)C} \odot \mathbf{d}^{(n-1)C+t}_{(n-1)C}) \quad (62)$$

$$\partial\mathbf{d}^{(n-1)C+(t-1)}_{(n-1)C} = (\partial\mathbf{A}^{'}_{(n-1)C+t} \odot \mathbf{a}_{(n-1)C+t})$$
$$- (\partial\mathbf{b}^{'}_{(n-1)C+t} \odot \mathbf{b}_{(n-1)C+t}) \oslash (\mathbf{d}^{(n-1)C+(t-1)}_{(n-1)C} \odot \mathbf{d}^{(n-1)C+(t-1)}_{(n-1)C}) \quad (63)$$

Given a vector $\mathbf{y}$, the relationship between $\partial\mathbf{y}$ and $\partial\log\mathbf{y}$ is:

$$\partial\log\mathbf{y} = \mathbf{y}(\partial\mathbf{y})$$

Thus:

$$\partial\log(\mathbf{d}^{(n-1)C+t}_{(n-1)C}) = (\partial\mathbf{q}^{'}_{[n]} \odot \mathbf{q}_{(n-1)C+t}) \odot \mathbf{d}^{(n-1)C+t}_{(n-1)C}$$
$$- (\partial\mathbf{k}^{'}_{(n-1)C+t} \odot \mathbf{k}_{(n-1)C+t}) \oslash (\mathbf{d}^{(n-1)C+t}_{(n-1)C}) \quad (64)$$

$$\partial\log\mathbf{d}^{(n-1)C+(t-1)}_{(n-1)C} = (\partial\mathbf{a}^{'}_{(n-1)C+t} \odot \mathbf{a}_{(n-1)C+t}) \odot \mathbf{d}^{(n-1)C+(t-1)}_{(n-1)C}$$
$$- (\partial\mathbf{b}^{'}_{[n]} \odot \mathbf{b}_{[n]}) \oslash (\mathbf{d}^{(n-1)C+(t-1)}_{(n-1)C}) \quad (65)$$

Notice that:

$$\partial\mathbf{q}^{'}_{(n-1)C+t} = \partial\mathbf{q}_{[n]} \oslash \mathbf{d}^{(n-1)C+t}_{(n-1)C} \quad (66)$$

$$\partial\mathbf{a}^{'}_{(n-1)C+t} = \partial\mathbf{a}_{(n-1)C+t} \oslash \mathbf{d}^{(n-1)C+(t-1)}_{(n-1)C} \quad (67)$$

$$\partial\mathbf{k}^{'}_{(n-1)C+t} = \partial\mathbf{k}_{(n-1)C+t} \odot \mathbf{d}^{(n-1)C+t}_{(n-1)C} \quad (68)$$

$$\partial\mathbf{b}^{'}_{(n-1)C+t} = \partial\mathbf{b}_{(n-1)C+t} \odot \mathbf{d}^{(n-1)C+t}_{(n-1)C} \quad (69)$$

$$\quad (70)$$

Substitude them into (eq. (64) and eq. (65)), the result is:

$$\partial\log(\mathbf{d}^{(n-1)C+t}_{(n-1)C}) = (\partial\mathbf{q}_{(n-1)C+t} \odot \mathbf{k}_{(n-1)C+t})$$
$$- (\partial\mathbf{k}_{(n-1)C+t} \odot \mathbf{k}_{(n-1)C+t}) - (\partial\mathbf{b}_{(n-1)C+t} \odot \mathbf{b}_{(n-1)C+t}) \quad (71)$$

$$\partial\log\mathbf{d}^{(n-1)C+(t-1)}_{(n-1)C} = (\partial\mathbf{a}_{(n-1)C+t} \odot \mathbf{a}_{(n-1)C+t}) \quad (72)$$

Review that $\mathbf{\Lambda}_{[n]}$ ($\tilde{\mathbf{\Lambda}}_{[n]}$) is the column-wise concatenation of $\mathbf{d}_{(n-1)C}^{(n-1)C+t}$ ($\mathbf{d}_{(n-1)C}^{(n-1)C+(t-1)}$) inside the $n$-th sequential chunk:

$$\mathbf{\Lambda}_{[n]} = \begin{bmatrix} \mathbf{d}_{(n-1)C}^{(n-1)C+1} & \mathbf{d}_{(n-1)C}^{(n-1)C+2} & \cdots & \mathbf{d}_{(n-1)C}^{nC} \end{bmatrix} \tag{73}$$

$$\tilde{\mathbf{\Lambda}}_{[n]} = \begin{bmatrix} \mathbf{d}_{(n-1)C}^{(n-1)C} & \mathbf{d}_{(n-1)C}^{(n-1)C+1} & \cdots & \mathbf{d}_{(n-1)C}^{nC-1} \end{bmatrix} \tag{74}$$

Therefore, the corresponding chunk-wise parallel forms for $\partial \log \mathbf{\Lambda}_{[n]}$ and $\partial \log \tilde{\mathbf{\Lambda}}_{[n]}$ are as follows:

$$\partial \log \mathbf{\Lambda}_{[n]} = \partial \mathbf{Q}_{[n]} \odot \mathbf{Q}_{[n]} - \partial \mathbf{K}_{[n]} \odot \mathbf{K}_{[n]} - \partial \mathbf{B}_{[n]} \odot \mathbf{B}_{[n]} \tag{75}$$

$$\partial \log \tilde{\mathbf{\Lambda}}_{[n]} = \partial \mathbf{A}_{[n]} \odot \mathbf{A}_{[n]} \tag{76}$$

For arbitrarily chosen $r_{ab}$ and $r_{kv}$, define the following $\mathrm{rankgather}_{r,C}$ operation. Suppose the operator matrix corresponds to "rank" $r$:

$$\mathbf{E}_{d \times C} = \mathrm{rankgather}_{r,C}(\mathbf{E}'_{d \times rC}) \tag{77}$$

Then $\mathbf{E}_{d \times C}$ is defined as follows:

$$\mathbf{E}_{:,t} = \sum_{i=0}^{r-1} \mathbf{E}'_{:,t*r+i} \tag{78}$$

Simply utilizing the above operator, we can extend eq. (75) and eq. (76) into rank-$r_{ab}$ low-rank term of decay with rank-$r_{kv}$ key-value updates:

$$\partial \log \mathbf{\Lambda}_{[n]} = \partial \mathbf{Q}_{[n]} \odot \mathbf{Q}_{[n]}$$
$$- \mathrm{rankgather}_{r_{kv},C}(\partial \mathbf{K}_{[n]} \odot \mathbf{K}_{[n]}) - \mathrm{rankgather}_{r_{ab},C}(\partial \mathbf{B}_{[n]} \odot \mathbf{B}_{[n]}) \tag{79}$$

$$\partial \log \tilde{\mathbf{\Lambda}}_{[n]} = \mathrm{rankgather}_{r_{ab},C}(\partial \mathbf{A}_{[n]} \odot \mathbf{A}_{[n]}) \tag{80}$$

## C.5 Deduction of HDLA's equivalent Test-Time Training formulae

First, let us revisit the optimization problem:

$$\mathbf{S}_{t,1} = \frac{\beta_t}{2} \min_{s_{t-1}} \|\mathbf{k}_t^\top \mathbf{S}_{t-1}\|^2, \tag{81}$$

$$\mathbf{S}_{t,2} = \min \left( \frac{1}{2} \mathrm{Trace}(\mathbf{S}_{t,1}^\top \mathrm{diag}(1 - \lambda_t) \mathbf{S}_{t,1}) \right), \tag{82}$$

$$\mathbf{S}_t = \frac{\beta_t}{2} \min_{s_{t,2}} \|\mathbf{k}_t^\top \mathbf{S}_{t,2} - \mathbf{v}_t^\top / \beta_t\|^2. \tag{83}$$

For the first optimization subproblem, online SGD yields:

$$\mathbf{S}_{t,1} = \mathbf{S}_{t-1} - \nabla_{\mathbf{S}_{t-1}} \left( \frac{\beta_t}{2} \min_{s_{t-1}} \|\mathbf{k}_t^\top \mathbf{S}_{t-1}\|^2 \right) \tag{84}$$

$$\tag{85}$$

$$= \mathbf{S}_{t-1} - \beta_t \mathbf{k}_t \mathbf{k}_t^\top \mathbf{S}_{t-1} \tag{86}$$

$$\tag{87}$$

$$= (\mathbf{I} - \beta_t \mathbf{k}_t \mathbf{k}_t^\top) \mathbf{S}_{t-1}. \tag{88}$$

For the second optimization subproblem, online SGD yields:

$$\mathbf{S}_{t,2} = \mathbf{S}_{t,1} - \nabla_{\mathbf{S}_{t,1}} \left( \frac{1}{2} \mathrm{Trace}(\mathbf{S}_{t,1}^\top \mathrm{diag}(1 - \lambda_t) \mathbf{S}_{t,1}) \right) \tag{89}$$

$$\tag{90}$$

$$= \mathbf{S}_{t,1} - (\mathbf{I} - \mathrm{diag}(\lambda_t)) \mathbf{S}_{t,1} \tag{91}$$

$$\tag{92}$$

$$= \mathrm{diag}(\lambda_t) \mathbf{S}_{t,1} \tag{93}$$

$$\tag{94}$$

$$= \mathrm{diag}(\lambda_t)(\mathbf{I} - \beta_t \mathbf{k}_t \mathbf{k}_t^\top) \mathbf{S}_{t-1}. \tag{95}$$

For the third optimization subproblem, online SGD yields:

$$\mathbf{S}_t = \mathbf{S}_{t,2} - \nabla_{\mathbf{S}_{t,2}} \left( \frac{\beta_t}{2} \min_{s_{t,2}} \| \mathbf{k}_t^\top \mathbf{S}_{t,2} - \mathbf{v}_t^\top / \beta_t \|^2 \right) \tag{96}$$

$$\tag{97}$$

$$= \mathbf{S}_{t,2} - \beta_t \mathbf{k}_t (\mathbf{k}_t^\top \mathbf{S}_{t,2} - \frac{1}{\beta_t} \mathbf{v}_t^\top) \tag{98}$$

$$\tag{99}$$

$$= (\mathbf{I} - \beta_t \mathbf{k}_t \mathbf{k}_t^\top) \mathbf{S}_{t,2} + \mathbf{k}_t \mathbf{v}_t^\top \tag{100}$$

$$\tag{101}$$

$$= (\mathbf{I} - \beta_t \mathbf{k}_t \mathbf{k}_t^\top) \mathrm{diag}(\lambda_t)(\mathbf{I} - \beta_t \mathbf{k}_t \mathbf{k}_t^\top) \mathbf{S}_{t-1} + \mathbf{k}_t \mathbf{v}_t^\top. \tag{102}$$

Thus, this three-step optimization problem yields the HDLA recursive formulation.

## D EXPERIMENTAL DETAILS

**Mechanistic Architecture Design.** In strict compliance with the MAD protocol, we employ a two layer token mixer-channel mixer architecture, where each layer's linear attention is aligned with a hidden state of dimension of 2048 (8 attention heads with $d_k = d_v = 16$), and run all experiments on NVIDIA H200 GPUs using bfloat16 precision. For state expansion experiments, the number of attention heads has been changed to 4.

**Zoology (Multi-Query Associative Recall).** The learning rates are swept by: np.logspace(-4, -2, 4) for sequence length 256, np.logspace(-5, -3, 4) for sequence length 512, [1e-5, 5e-5, 1e-4, 5e-4, 1e-3, 5e-3, 1e-2] for sequence length 1024 and 2048, and we take the best result from all learning rates. The parameter scales are aligned to 1.65M.

**Language Modeling.** We trained models on fineweb-edu, including small-scale and large-scale versions. For the small-scale version, we trained for 10B tokens with a learning rate of 3e-4, sequence length (seqlen) of 2048, and a total batch size of 256 (num gpu × batch per gpu × grad acc). For the large-scale version, we trained for 100B tokens with a learning rate of 3e-4, seqlen=8k, and a total batch size of 128. All experiments were conducted on 8/32 A100 GPUs. We used FLA to implement the model, the Flame framework for training, and lm-eval-harness for evaluation. We report results on wikitext, lambada_openai, boolq, piqa, hellaswag, winogrande, arc_easy, arc_challenge, openbookqa, social_iqa, swde, squad_completion, and fda. For wikitext (word_perplexity) and lambada_openai (perplexity), as well as for swde, squad_completion, and fda, we report the Exact-Match (EM) score; for hellaswag, arc_challenge, and openbookqa, we report acc_norm; for the rest, we report accuracy (acc).

**Image Classification.** Each model is trained and evaluated on 4 NVIDIA A800 GPUs using Pytorch DDP. The input size of ImageNet is $224 \times 224$. Following Deit, the batch size is set to 2048 during 300 training epochs with a cosine decay learning rate whose peak value is $2.4 \times 10^{-3}$. The warmup

epochs is set to 20. We choose AdamW ($\beta_1 = 0.9$, $\beta_2 = 0.98$) with $0.05$ weight decay as the optimizer. Note that we do not use cutmix or mixup during the training. Results of MetaLA —Chou et al. (2024), HGRN Qin et al. (2023), GLA Yang et al. (2024a), Mamba Gu & Dao (2024) and Deit Touvron et al. (2021) are directly borrowed from Chou et al. (2024).

