# OpenReview forum: "Householder-Diagonalized Linear Attention (HDLA): Utilizing Enhanced Decay Mechanism for Efficient Sequence Modeling"
_ICLR.cc/2026/Conference — ICLR 2026 Poster_

### Official Review · Reviewer_Pma4 · 2025-10-27

**Soundness:** 3
**Presentation:** 2
**Contribution:** 3
**Rating:** 8
**Confidence:** 4

**Summary:**

The authors propose a diagonal-plus-rank-2 decay / state-transition matrix for linear attention/SSM by using products of householder matrices and diagonal matrices. They also provide a chunk-wise parallel algorithm for this sequential recurrence as a generalization of previous versions with rank 1 constraint. In addition, the authors show that the proposed HDLA method can be viewed as performing online update to the recurrence hidden state, which is a form of “test-time training”. The proposed method achieves good performance on MAD synthetics, MQAR recall tasks, language modeling perplexity and downstream reasoning tasks, retrieval-based S-NIAH tasks, and image classifications.

**Strengths:**

- The proposed method combines lots of prior efforts (DeltaProduct, Gated Delta Net, etc) to continuously improve the linear attention family of models. This is a good contribution to this line of work.
- The derivation of the chunking algorithm is quite involved but definitely useful for the community.
- Empirical experiments are extensive

**Weaknesses:**

It’s not immediately obvious how the newly proposed decay matrix is a form of diagonal-plus-rank-2 update. It’s better to lay this out more clearly instead of leaving it to the reader. For general machine learning audience, this is not that obvious so it should be reflected more in the writing.

**Questions:**

- It’s good to impose new structured assumption in the decay matrix P_t for balancing between efficiency and expressivity, and the chunking algorithm also alleviate some of the computational overhead. However, the proposed HDLA includes a lot more matrix multiplications. In fact, any structured matrix which is formed by a composition of permutation, block-diagonal assumption, summations etc can be viewed as a deep linear transformations which causes more I/O due to sequential matrix loading, computation, and writing between HBM and SRAM. How do you optimize the I/O efficiency here?
- By using a diagonal-plus-2-rank update, you allocate more compute for the decay matrix. But the underlying hidden state matrix S_t is still finite and will still suffer from retrieval failures under long sequence input. What if you allocate that compute not to the decay matrix P_t but to have basically a bit more S_t, i.e. perhaps allocating a few smaller S_t matrix across the sequence dimension. What do you think would be superior here?

---

> ### Author Response · Authors · 2025-11-26
> **(1/2) Response to Reviewer Pma4 (Questions 1-2)**
>
> Dear Reviewer Pma4,
>
> We sincerely thank you for your devoted review. We have listed your concerns point by point and provided our responses. If there is anything we have not addressed sufficiently, please let us know—we would be very happy to have a further discussion with you.
>
> ****
> **Q1**: Why HDLA's decay can form a Diagonal-Plus-Rank-2 structure.
> ****
>
> **A**: We've provided the process of transforming the HDLA decay matrix into a Diagonal-Plus-Rank-2 structure in lines 792–827 (Appendix C.2). Let's present a more detailed explanation here.
>
> Consider factorizing each element of $\mathbf{\Lambda}_t$ as the product of its square roots, then coupling them with the left and right Householder transformations $\mathbf{H}_t$. This yields the reformulation of $\mathbf{P}_t$ as follows:
>
> $
>     \mathbf{P}_t = \left(\mathrm{Diag}(\sqrt{\lambda_t}) - \beta_t\mathbf{k}_t(\mathbf{k}_t\odot \sqrt{\lambda_t})^\top\right)\left(\mathrm{Diag}(\sqrt{\lambda_t}) - \beta_t(\mathbf{k}_t\odot \sqrt{\lambda_t})\mathbf{k}_t^\top\right)
> $
>
> This form of $\mathbf{P}\_t$ is a special case of two diagonal-plus-rank-one matrices' product:
>
> $
>     \mathbf{P}\_t = (\mathbf{D}\_{t, (1)} - \mathbf{a}\_{t, (1)}\mathbf{b}\_{t, (1)}^\top)(\mathbf{D}\_{t, (2)} - \mathbf{a}\_{t, (2)}\mathbf{b}\_{t, (2)}^\top)
> $
>
> where:
>
> $\mathbf D_{t, (1)} = \mathbf D_{t, (2)} = \mathrm{Diag}(\sqrt \lambda_t),
> \mathbf a_{t, (1)} = \beta_t \mathbf k_t,
> \mathbf a_{t, 2} = \beta_t (\mathbf k_t \odot \sqrt{\lambda_t}),
> \mathbf b_{t, (1)} =  (\mathbf k_t \odot \sqrt{\lambda_t}),
> \mathbf b_{t, 2} = \mathbf k_t$
>
> We demonstrate the detailed formulation of $\mathbf{A}_t, \mathbf{B}_t$ as follows:
>
> $
>     \mathbf{P}\_t = (\mathbf{D}\_{t, (1)} - \mathbf{a}\_{t, (1)}\mathbf{b}\_{t, (1)}^\top)(\mathbf{D}\_{t, (2)} - \mathbf{a}\_{t, (2)}\mathbf{b}\_{t, (2)}^\top)\notag \\
>     = \mathbf{D}\_{t, (1)}\mathbf{D}\_{t, (2)} - \mathbf{a}\_{t, (1)}\mathbf{b}\_{t, (1)}^\top\mathbf{D}\_{t, (2)} - (\mathbf{D}\_{t, (1)} - \mathbf{a}\_{t, (1)}\mathbf{b}\_{t, (1)}^\top)\mathbf{a}\_{t, (2)}\mathbf{b}\_{t, (2)}^\top \notag
> $
>
> Let:
>
> $
>     \mathbf{D}\_t = \mathbf{D}\_{t, (1)}\mathbf{D}\_{t, (2)} \in \mathbb{R}^{d\_k \times d\_k} \\
>     \mathbf{A}\_t = \begin{bmatrix}\mathbf{a}\_{t, (1)} & (\mathbf{D}\_{t, (1)} - \mathbf{a}\_{t, (1)}\mathbf{b}\_{t, (1)}^\top)\mathbf{a}\_{t, (2)} \end{bmatrix} \in \mathbb{R}^{d\_k \times 2} \\
>     \mathbf{B}\_t = \begin{bmatrix}\mathbf{D}\_{t, (2)}\mathbf{b}\_{t, (1)} & \mathbf{b}\_{t, (2)}\end{bmatrix} \in \mathbb{R}^{d\_k \times 2}
> $
>
> Then $\mathbf{P}_t$ can be rewritten as the following diagonal-plus-rank-$2$ form:
>
> $
>     \mathbf{P}_t = \mathbf{D}_t - \mathbf{A}_t\mathbf{B}_t^\top \in \mathbb{R}^{d_k \times d_k}, \mathbf{A}_t, \mathbf{B}_t \in \mathbb{R}^{d_k \times 2}
> $
>
> ****
> **Q2: HDLA's I/O efficiency optimization methods.**
> ****
>
> **A**:
>
> **(1) I/O Optimization Methods for HDLA's Chunk-wise Parallel Kernel.**
>
> Our current implementation of the HDLA chunked parallel kernel is as follows:
> - We first use Torch code to pre-construct the inputs $\mathbf A_t, \mathbf B_t$ corresponding to Equation $(7)$.
> - Then, we use a Triton kernel to compute the WY Representation and write it back to HBM (High Bandwidth Memory).
> - Finally, we load the WY Representation for linear attention computation and hidden state update.
>
> Therefore, with our current approach of writing a general Triton kernel for $r_{ab} = 2, r_{kv} = 1$, there are some intermediate results that require I/O operations between HBM and SRAM, and the I/O volume is related to both $r_{kv}$ and $r_{ab}$. This constitutes the main limitation of our current general implementation in terms of I/O efficiency.
>
> (1.2) To reduce the I/O overhead of HDLA, we can consider kernel fusion in future implementations.
>
> For example, after fusing the kernels for $A_t, B_t$ computations and WY Representation computation, we only need to load the necessary vectors (i.e., $\mathbf{k}_t, \mathbf{\lambda}_t, \mathbf{\beta}_t$, etc), thus avoid the I/O overhead caused by writing $\mathbf{A}_t$ and $\mathbf{B}_t$ to HBM and then reading them.
>
> **(2) I/O Optimization Methods for HDLA Recurrent Decode Kernel.**
>
> When implementing HDLA's recurrent decode kernel, we have already adopted kernel fusion, i.e., all linear attention operations involved in HDLA (including the construction of the decay matrix as in Equation $(5)$, the hidden state update as in Equation $(8)$, and the linear attention output computation as in Equation $(4)$) are fused into a single Triton kernel.
>
> The benefit of this approach is a significant reduction in I/O volume: Only $\mathbf{q}\_t, \mathbf{k}\_t, \mathbf{v}\_t, \mathbf{S}\_{t-1}, \mathbf{\beta}\_t, \mathbf{\lambda}\_t$ need to be loaded, and only $\mathbf{S}\_{t}$ and $\mathbf{o}\_t$ are written back. This I/O volume is almost on par with GLA, and the throughput of our inference kernel reaches about 89% of GLA (see our **Public Comment 3** for details).

---

> ### Author Response · Authors · 2025-11-26
> **(2/2) Response to Reviewer Pma4 (Question 3)**
>
> ****
> **Q3: Whether expanding the scale of S_t or complicating the design of P_t is superior.**
> ****
>
> **A**: In fact, increasing the size of $S_t$ and making the structure of $P_t$ more complex are **two orthogonal directions**, and **their benefits can be combined**. **We believe both approaches are equally important and can each bring significant advantages.**
>
> **(1) Increasing the complexity of the $P_t$ structure.**
> - **Leveraging diagonal-plus-rank-2 structured decay matrix, HDLA consistently achieves notable improvements** on tasks such as RULER and Commonsense Reasoning, **compared to** previous baselines with **diagonal-plus-rank-1 decay**.
> - This year (2025), there has also been **a clear industry trend toward continually increasing the complexity of $P_t$ within linear attention–softmax attention hybrid architectures, which fully aligns with the observation in the INTRODUCTION of this paper.**. Here we list three examples:
>
> | Model | Release Time | Decay Structure of Linear Attention|Description|
> | :---: | :---: | :---: |:---:|
> |Minimax-01[4]| 2025.01|$\alpha \mathbf I$ | Diagonal, constant, scalar decay|
> |Qwen3-Next[5]| 2025.09|$\alpha_t(\mathbf I - \beta_t \mathbf k_t\mathbf k_t^\top), \alpha_t \in \mathbb R$|Diagonal-Plus-Rank-1 decay with scalar diagonal term|
> |Kimi Linear[6]| 2025.10| $(\mathbf I - \beta_t \mathbf k_t\mathbf k_t^\top) \mathrm{Diag}(\mathbf d_t), \mathbf d_t \in \mathbb R^{d_k}$|Diagonal-Plus-Rank-1 decay with vector diagonal term|
>
>
> **(2) Expanding the size of $S_t$**. This approach offers at least three key advantages:
> - It **fundamentally increases memory capacity** by multiples.
> - It allows for **specialization across different memory units**.
>     - For example, Table 5 in the MoM[2] paper shows that when the hidden state is scaled up and sparsely activated, different hidden state units tend to specialize in remembering different types of tokens (e.g., basic verbs and nouns, scientific or technical terms), and in achieving different objectives (such as capturing basic semantic knowledge, specialized knowledge, or detailed information).
>
> References: \
> [1] https://arxiv.org/pdf/2507.01004 \
> [2] https://arxiv.org/pdf/2502.13685 \
> [3] https://arxiv.org/abs/2507.16577

---

> > ### Comment · Reviewer_Pma4 · 2025-11-27
> >
> > Thanks for following up on my questions! I maintain my score.

---

### Official Review · Reviewer_BfPn · 2025-10-29

**Soundness:** 3
**Presentation:** 2
**Contribution:** 2
**Rating:** 4
**Confidence:** 3

**Summary:**

The paper proposes HDLA which is a linear attention variant that parameterizes a decay matrix as a diagonal matrix with two structured factors computed through generalized Householder matrices. This HDLA attention mechanism performs relative well at a small scale when compared to softmax attention and better than appropriate baselines.

**Strengths:**

* I find sound and comprehensive the efficiency constraints laid out in section 3.1.1.
* The empirical results compared to different variants of the Gated DeltaProduct are strong and comprehensive.
* I appreciate the connections of HDLA from the TTT perspective.

**Weaknesses:**

* The community's experience with subquadratic alternatives to softmax attention is that you can find efficiency benefits
for small models (around 7B params) but these benefits fade away as we increase the scale.
Does your paper advocate to try HDLA for LLMs? Or are you content with the results at the < 3B scale?
I understand that as a researchers we might not posses the access to compute to train such large models,
but is there any indication that HDLA would not suffer diminishing returns at larger scales?

* It is unclear to me how can I incorporate HDLA into a transformer right now. The equations are presented for each
element in the sequence, which is great for the analysis, but then it is unclear how to incorporate this
method in practice and what implementation details need to be considered.

* There are no runtime comparisons which makes it harder to understand the overhead of HDLA against baselines like DeltaProduct.

**Questions:**

* Why does criteria P2 presents $H_t$ as an invertible transformation instead of a orthonormal basis? Indeed, $H_t$ is
invertible when it is an orthonormal basis but it is a stronger result from the spectral theorem that you can find such
an $H_t$ for a real symmetric matrix. Moreover, the construction in equation 5 requires orthonormality. Please help me
see if I'm missing a detail.

* I didn't follow the three-step optimization process in equations 19-21 (lines 309-318). I only see an optimization
problem in Eq. 19, the next two lines appear to be just updates. Am I missing something?
Is there any reason as to why you didn't shared code through an anonymized link? I'm wondering how you implemented HDLA
and its chunk-wise parallel version. Especially as you emphasize the I/O cost which to optimize it would require custom
kernels.

* In Table 3, why is Mamba considered a linear attention baseline? Does Mamba not scale as $\mathcal{O}(N \log N)$?, where
$N$ is the sequence length.

* Could you provide runtime comparisons of HDLA for training and inference? Some examples suffice to get an estimate.

---

> ### Author Response · Authors · 2025-11-26
> **(1/2) Response to Reviewer BfPn (Questions 1-4)**
>
> Dear Reviewer BfPn,
>
> We sincerely thank you for your devoted review. We have listed your concerns point by point and provided our responses. If there is anything we have not addressed sufficiently, please let us know—we would be very happy to have a further discussion with you.
>
> ****
> **Q1: Whether to make attempts to train larger-scale language models using HDLA, and the efficiency degradation problem when scaling up.**
> ****
>
> **A**: **(1) We are very willing to scale up HDLA (in the form of the HDLA–Softmax Attention hybrid architecture) in future work.**
>
> This year, three major open-source large language models in the industry have adopted increasingly sophisticated parameterization schemes for the decay matrix in linear attention–softmax attention hybrid architectures. **This trend aligns with the observations discussed in the INTRODUCTION section of our paper, and supports the observation that further enhancing the complexity of the decay matrix can elevate the upper bound of linear attention mechanisms.**
>
> - The Minimax-01[1] hybrid model, released in January 2025, employs Lightning Attention as its linear attention backbone. Its decay matrix has the form $\alpha \mathbf{I}$, representing a diagonal, constant, scalar decay.
> - The Qwen3-Next[2] hybrid model (80B total parameters, 3B activated parameters), launched in September 2025, utilizes Gated DeltaNet as its linear attention backbone. Its decay matrix is structured as $\alpha_t (\mathbf{I} - \beta_t \mathbf{k}_t \mathbf{k}_t^\top)$, which is a typical diagonal-plus-rank-1 structure.
> - The Kimi Linear hybrid model (48B total parameters, 3B activated parameters), introduced in October 2025, leverages Kimi Delta Attention as its linear attention backbone. Its decay matrix takes the form $(\mathbf{I} - \beta_t \mathbf{k}_t \mathbf{k}_t^\top)\text{Diag}(\lambda_t)$, a diagonal-plus-rank-1 structure that replaces the scalar diagonal terms of the Gated DeltaNet decay matrix with vectors.
>
> **(2) Our ideas on addressing the "efficiency degradation problem" of HDLA when scaling up.**
>
> **(2.1) Whether the efficiency benefits of linear attention will degradate when paramater scale >7B**
>
> **Based on our knowledge, the efficiency advantage of linear attention remains significant at scales above 7B parameters.**
> - For example, Kimi-Linear[4] (which adopts a KDA linear attention–softmax attention 3:1 hybrid architecture) achieves 1.16 times the computational efficiency of pure multi-head latent attention with a total parameter count of 48B.
> - Similarly, the Qwen3-Next-80B-A3B[3] model uses a Gated DeltaNet linear attention–softmax attention 3:1 hybrid architecture, and with a total parameter scale of 80B, it is 2–3 times faster than Qwen3-30B-A3B (which uses a group query softmax attention architecture and has 30B parameters).
>
> **Although HDLA is somewhat slower than the baseline, its linear complexity remains unchanged. Therefore, when scaling up, HDLA should still have a fundamental efficiency advantage over softmax attention.**
>
> (2.2) To further enhance the efficiency of HDLA during scale-up, we can consider implementing custom operators via Triton kernel fusion, which reducies the I/O cost of intermediate results.
>
> ****
> **Q2: Implementation details to consider for HDLA, the custom kernels of HDLA, and how to integrate it into Transformers.**
> ****
>
> **A**: Please refer to our **Public Comment 1** for details.
>
> ****
> **Q3: Comparison of training and inference time between HDLA and other baselines.**
> ****
>
> **A**: Please refer to our **Public Comment 2 & 3** for details.
>
> ****
> **Q4: The time complexity of Mamba.**
> ****
>
> **A**: The time complexity of Mamba is exactly $O(N)$, where $N$ represents the sequence length.
> - According to Table 1 in [9], Mamba can be identified as a special parameterization of the unified linear attention  proposed in [9] (Equations $(11)-(18)$), with input-dependent diagonal decay term.
> - Therefore, Mamba can be computed using the FlashLinearAttention algorithm [10] (see Algorithms 3 and 4 in the paper), which has strictly linear time complexity.

---

> ### Author Response · Authors · 2025-11-26
> **(2/2) Response to Reviewer BfPn (Questions 5-6)**
>
> ****
> **Q5: Mathematical issues related to diagonalization and H_t.**
> ****
>
> **A**: **(1) The reason why Criterion $(P2)$ refers to $H_t = I - \beta_t k_tk_t^\top$ as an invertible transformation rather than an orthogonal one.**
>
> In fact, the eigenvalues of $H_t$ are $1$ (with multiplicity $d_k - 1$) and $1 - \beta_t \| k_t \|^2$ (with multiplicity $1$).
> - As long as $1 - \beta_t \| k_t \|^2$ is nonzero, $H_t$ is invertible.
> - However, due to $1 - \beta_t \| k_t \|^2$, $H_t$ is not orthogonal.
>
> The orthogonal Householder matrix typically refers to the special case where $\beta_t = 2$, while the generalized Householder matrix here adopts a more flexible range, allowing $\beta_t \in (0, 1)$ or $\beta_t \in (0, 2)$.
>
> **(2) Using non-orthogonal generalized Householder matrices instead of strictly orthogonal Householder matrices is a common practice in linear attention (e.g., DeltaNet, GDN, and GDP all adopt this approach).**
>
> From the perspective of Test-Time Training, $\beta_t$ represents the learning rate for value prediction and gradient descent at the current timestep (see Equation $(21)$). An input-dependent, flexible learning rate $\beta_t \in (0, 2)$ (corresponding to a generalized Householder matrix), rather than a constant learning rate of $2$ (corresponding to a standard Householder matrix), is a common choice in linear attention.
>
> **(3) Whether the construction of Equation (5) requires orthogonality.**
>
> In fact, according to the definition of congruence transformations [6], as long as there exists an invertible matrix $\mathbf P_t$ such that $\mathbf H_t = \mathbf P_t \mathbf \Lambda_t \mathbf P_t^{\top}$, the matrices $\mathbf H_t$ and $\mathbf \Lambda_t$ are congruent. This definition places no strict orthogonality requirement on the invertible matrix $\mathbf P_t$.
>
> ****
> **Q6: The three-step optimization in formulas (19) to (21).**
> ****
>
> **A**: **We agree with your understanding. Equations $(20)$ and $(21)$ together constitute exactly the Gated DeltaNet-style test-time training learning rule, while Equation $(19)$ is a distinctive feature of HDLA that differentiates it from models such as Gated DeltaNet and DeltaNet.**
>
> Let us provide a detailed explanation of each equation for you.
>
> **(1) Equation $(19)$ can be regarded as a means of proactively eliminating redundant information in $S_{t-1}$.**
> - $k_t^\top S_{t-1}$ means **computing the inner product similarity between $k_t$ and each column of $S_{t-1}$**.
> - **Optimizing the resulting vector towards the zero vector** can be seen as: **Proactively eliminating some redundant information in the historical state $S_{t-1}$, which are related to the current key $k_t$**, before performing hidden state update using $k_tv_t^\top$.
> - This optimization improves the utilization of $S_{t-1}$'s limited capacity, and is also one explanation for why HDLA demonstrates significantly stronger retrieval capabilities than Gated DeltaNet in Table 6.
>
>
> $
> \mathbf S_{t, 1}
> =\mathbf S_{t-1}- \nabla_{\mathbf S_{t-1}}\left(\frac {\beta_t} {2} \min_{s_{t-1}} \| \mathbf k_t^\top \mathbf S_{t-1} \|^2 \right) \\\\
> = \mathbf S_{t-1} - \beta_t \mathbf k_t \mathbf k_t^\top \mathbf S_{t-1} \\\\
> = (\mathbf I - \beta_t \mathbf k_t \mathbf k_t^\top) \mathbf S_{t-1}.
> $
>
> (2) Formula $(20)$ adopts a regularization term similar to MesaNet [7], performing "weight decay" on $|S_{t, 1}^\top \mathrm{Diag}(1 - \lambda_t) S_{t, 1} |$.
>
> $\mathbf S_{t,2} = \mathbf S_{t, 1} - \nabla_{\mathbf S_{t, 1}}
> \left(
> \frac 1 2 \mathrm{Trace}(\mathbf S_{t,1}^\top \mathrm{diag}(1- \lambda_t) \mathbf S_{t,1})\right) \\\\
> = \mathbf S_{t, 1} - (\mathbf I - \mathrm{diag}(\lambda_t))\mathbf S_{t, 1} \\\\
> = \mathrm{diag}(\lambda_t)\mathbf S_{t, 1} \\\\
> = \mathrm{diag}(\lambda_t)(\mathbf I - \beta_t \mathbf k_t \mathbf k_t^\top) \mathbf S_{t-1}$
>
> (3) Formula $(21)$ exactly corresponds to the value prediction and gradient descent operation in Delta Rule [8]:
>
> $
> \mathbf S_{t} = \mathbf S_{t, 2} - \nabla_{\mathbf S_{t,2}}
> \left(
> \frac{\beta_t}{2} \min_{s_{t, 2}} \|\mathbf k_t^\top \mathbf S_{t,2} - \mathbf v_t^\top/\beta_t\|^2\right) \\\\
> = \mathbf S_{t, 2} - \beta_t \mathbf k_t(\mathbf k_t^\top \mathbf S_{t,2} - \frac{1}{\beta_t}\mathbf v_t^\top) \\\\
> = (\mathbf I - \beta_t \mathbf k_t \mathbf k_t^\top )\mathbf S_{t, 2} + \mathbf k_t \mathbf v_t^\top \\\\
> = (\mathbf I - \beta_t \mathbf k_t \mathbf k_t^\top )\mathrm{diag}(\lambda_t)(\mathbf I - \beta_t \mathbf k_t \mathbf k_t^\top) \mathbf S_{t-1} + \mathbf k_t \mathbf v_t^\top.
> $
>
> References: \
> [1] https://arxiv.org/abs/2312.00752 \
> [2] https://arxiv.org/abs/2501.08313 \
> [3] https://qwen.ai/blog?id=4074cca80393150c248e508aa62983f9cb7d27cd&from=research.latest-advancements-list \
> [4] https://arxiv.org/abs/2510.26692 \
> [5] https://arxiv.org/abs/2507.01004 \
> [6] https://en.wikipedia.org/wiki/Matrix_congruence \
> [7] https://arxiv.org/abs/2506.05233v1 \
> [8] https://arxiv.org/abs/2406.06484 \
> [9] https://arxiv.org/pdf/2411.10741 \
> [10] https://arxiv.org/pdf/2312.06635

---

### Official Review · Reviewer_aNs6 · 2025-11-01

**Soundness:** 3
**Presentation:** 2
**Contribution:** 3
**Rating:** 6
**Confidence:** 3

**Summary:**

This paper proposes HDLA, a linear attention mechanism that parameterizes the decay matrix with a structured Diagonal-plus-Rank-2 form. This design aims to improve expressivity over the common Diagonal-plus-Rank-1 family while maintaining parameter, memory, and computational efficiency. The authors also derive a rank-generalized chunk-wise parallel algorithm that supports arbitrary Diagonal-plus-Rank-$r_{ab}$ decay and rank-$r_{kv}$ key–value updates, using a WY-style representation and custom block-triangular operators to enable efficient intra-/inter-chunk computations. Empirically, HDLA outperforms strong linear-attention baselines on language modeling (0.4B/1.45B/2.8B), synthetic benchmarks (MAD, MQAR), and retrieval tasks (RULER), with particularly large gains on MQAR and RULER. The paper also provides a Test-Time Training interpretation of the update rule and reports competitive ImageNet classification performance. Limitations are discussed (recency bias, need for state expansion and potential non-linear updates).

**Strengths:**

- **Principled architectural design**: The authors motivate and provide a principled, structured Diagonal-plus-Rank-2 decay that extends rank-1 methods while preserving parameter efficiency (O(dk)).
- **Generalized chunk-wise parallel algorithm**: The authors also present a unifying "WY-"style framework accommodating Diagonal-plus-Rank-$r_{ab}$ decay and rank-$r_{kv}$ KV updates, which subsumes several prior works as special cases.
- **Strong and diverse empirical results**: The architecture shows consistent improvements over strong linear-attention baselines (GLA, Mamba2/HGRN2, DeltaNet/Gated DeltaNet, GDP2/3) across LM perplexity at multiple scales, synthetics such as MQAR and MAD, and longer context RULER tasks.

**Weaknesses:**

- **Presentation**: Some instances could be improved with a bit more detail.
  - For example, the "WY Representation" should be defined or at least alluded to, e.g., with a reference to [1].
  - Separately, while I appreciate the authors' acknowledged comparison to softmax attention, it would be good to see the delta between HDLA and softmax attention on the retrieval tasks such as NIAH or MQAR (i missed these in Table 6 and Figure 2).
  - To present the advantage of the more expressive Rank-2 structure, could the authors also present a results table (or expand Table 1) that puts side-by-side the model, it's expressivity, and the average empirical performance binned by evaluation (ppl, retrieval acc, LM-eval) ?
  - Please also fix the first citation: "Sanjeev Arora, Bowen Yang, Selim Eyuboglu, Aditya Narayan, Alexis Hojel, Imanol Trummer, and Christopher Re." fwiw in a random sample of a couple other citations they all seemed to correspond to real author lists.

- **Wall-clock / "Real World" efficiency**: While I appreciated the complexity analysis and inclusion of efficient algorithms (e.g., chunk-wise parallel), and the claims mostly center on presenting a new decay structure + evaluating it's modeling quality, in being a work on linear attention it would be good to validate how the efficiency translates to real world efficiency boosts, e.g., by measuring wall-clock inference time or generation throughput.

[1]  Christian Bischof and Charles Van Loan. The WY Representation for Products of Householder Matrices. 1985.

**Questions:**

Please see Weaknesses above.

---

> ### Author Response · Authors · 2025-11-26
> **(1/2) Response to Reviewer aNs6 (Questions 1-3)**
>
> Dear Reviewer aNs6,
>
> We sincerely thank you for your devoted review. We have listed your concerns point by point and provided our responses. If there is anything we have not addressed sufficiently, please let us know—we would be very happy to have a further discussion with you.
>
> ****
> **Q1: Comparison of wall clock/real-world efficiency between HDLA and other models.**
> ****
>
> **A**: Please refer to our Public Comment 2 and Public Comment 3 for details.
>
> ****
> **Q2: Detailed introduction to WY Representation.**
> ****
>
> **A**: WY Representation is a compact mathematical framework for efficiently computing the cumulative product of a certain class of structured matrices, namely DPLR (Diagonal Plus Low-Rank) matrices. By observing and reformulating the original form of the cumulative product of DPLR matrices, it eliminates the $\sum$ and $\prod$ operators that are dependent on the number of cumulative products, thus enabling the cumulative product of an arbitrary number of DPLR matrices to be completed via concise matrix-matrix operations.
>
> **Since the derivation of WY Representation in this paper fully covers the derivation processes in previous works, we only present here the different types of matrix cumulative products for which WY Representation was proposed in each work.**
> - Christian Bischof and Charles Van Loan [2]  first proposed the original form of WY Representation to address the problem of cumulative products of Householder matrices, i.e., $\mathbf I - 2 \mathbf k_t\mathbf k_t^\top, \mathbf k_t \in \mathbb R^{d_k \times 1}$.
> - The DeltaNet Paper[3] extended the original WY Representation to handle cumulative products of Identity-Plus-Rank-1 matrices, i.e., $\mathbf I -  \mathbf a_t\mathbf b_t^\top, \mathbf a_t, \mathbf b_t \in \mathbb R^{d_k \times 1}$.
> - A technical documentation [4] in the FlashLinearAttention repository proposed an efficient computational form for the cumulative product of more general Diagonal-Plus-Rank-1 matrices, i.e., $\mathrm {Diag}(\lambda_t) -  \mathbf a_t\mathbf b_t^\top, \mathbf a_t, \mathbf b_t, \lambda_t \in \mathbb R^{d_k \times 1}, \beta_t \in (0, 2)$.
> - This paper’s WY Representation further expands the rank of the low-rank term in Diagonal-Plus-Low-Rank matrices, allowing the cumulative product of diagonal plus rank-$r_{ab}$matrices, i.e., $\mathrm Diag (\lambda_t) - \mathbf A_t \mathbf B_t^\top, \mathbf A_t, \mathbf B_t \in \mathbb R^{d_k \times r_{ab}}, \lambda_t \in \mathbb R^{d_k \times 1}$,  to be efficiently computed.
>
> **A more formal and comprehensive introduction to WY Representation will be fully presented in the updated version of the PDF.**
>
> ****
> **Q3: Performance of Softmax Attention on MQAR and NIAH tasks.**
> ****
>
> **A**: **(1) The MQAR results of Softmax Attention**
>
> |Sequence Length| Accuracy |
> |:-:|:-:|
> |256|99%|
> |512|99%|
> |1024|100%|
> |2048|99%|
>
> - When the training length is equal to the test length (i.e., without extrapolation), Softmax Attention is almost fully capable of handling the various MQAR scenarios we designed.
>
> (2) RULER
> - We apologize for not being able to complete the RULER experiments at this time. Due to limitations in computational resources, we have not yet been able to further pretrain the 1.45B-parameter Llama model on 50B tokens, which is necessary to produce fully fair experimental results comparable to Gated DeltaNet and HDLA in Table 6.
> - The general conclusion is that, without extrapolation (i.e., inference length not exceeding the training length), softmax attention can achieve close to 100% accuracy on RULER tasks that are not too long.
> - For now, **we cite recent work for the evaluation of Softmax Attention on the RULER task.** The first row of Table 4 in [1] presents the performance of Softmax Attention (420M parameters, 7B tokens, 2K training length) on RULER. When the evaluation lengths are 1K and 2K, the accuracy of Softmax Attention is close to 100%; however, when the evaluation length exceeds the training length (i.e., 4K and 8K), the accuracy drops significantly:
>
> |Seqlen (S-NIAH-1) | Accuracy | Seqlen (S-NIAH-2)  | Accuracy |
> |:-|:-|:-|:-|
> | 1K            | 100.0         | 1K            | 100.0         |
> | 2K            | 99.4          | 2K            | 100.0         |
> | 4K            | 94.2          | 4K            | 4.8        |
> | 8K            | 11.4          | 8K            | 0.0          |
> | 16K           | 0.8           | 16K           | -          |

---

> ### Author Response · Authors · 2025-11-26
> **(2/2) Response to Reviewer aNs6 (Questions 4-5)**
>
> ****
> **Q4: A table for comprehensive comparison of overall performance and expressivity.**
> ****
>
> **A**: According to your suggestion, we create the following table to provide a comprehensive comparison of each model’s expressiveness, language modeling perplexity (PPL), common-sense reasoning (CSR), and retrieval (RET) abilities. This table will be included in the revised PDF.
>
> |Model|Expressiveness|PPL-0.4B|CSR-0.4B|RET-0.4B|PPL-1.45B|CSR-1.45B|RET-1.45B|PPL-2.8B|CSR-2.8B|RET-2.8B|
> |:-:|:-:|:-:|:-:|:-:|:-:|:-:|:-:|:-:|:-:|:-:|
> |HDLA|Diagonal-Plus-Rank-2 structured decay with generalized Householder diagonalization|**36.06**|**47.46**|11.71|**22.32**|**51.06**|19.36|**18.58**|**53.09**|*24.87*|
> |GDN|Diagonal-Plus-Rank-1 Decay with scalar diagonal term|43.06|46.65|10.47|24.83|*50.80*|18.46|*19.60*|*52.67*|24.66|
> |GDP2|Performs 2 GDN iterations in a timestep|41.28|46.97|11.79|24.65|49.61|17.60|20.38|52.07|22.09|
> |GDP3|Performs 3 GDN iterations in a timestep|46.22|46.69|10.80|26.80|49.32|15.21|-|-|-|
> |DN|Identity-Plus-Rank-1 Decay|44.54|46.33|13.32|27.44|49.87|19.68|22.69|51.33|23.96|
> |HGRN2|Diagonal decay (input-dependent, vector) with increased lower bound from bottom layers to upper ones|39.34|*47.15*|10.73|23.98|50.28|18.39|20.31|52.35|24.08|
> |Mamba2|Diagonal decay (input-dependent, scalar)|40.63|46.23|*15.38*|25.73|49.51|*20.33*|22.78|51.01|22.49|
> |GLA|Diagonal decay (input-dependent, vector)|43.75|47.08|8.98|26.42|50.58|14.98|21.45|-|-|
> |TNL|Diagonal decay (input-independent, scalar)|41.25|47.14|11.13|26.28|49.93|16.59|21.72|51.64|20.42|
> |Llama|Softmax Attention|*37.60*|47.14|**31.69**|*23.68*|50.39|**46.71**|20.71|51.84|**46.27**|
>
> Since RULER only evaluates two models, and the experiments on MAD, MQAR, and image classification are on a relatively small scale, these results are not included in the table.
>
> ****
> **Q5: Citation errors.**
> ****
>
> **A**: We apologize for the errors in the authors' names and sincerely thank you for pointing them out. In addition to the incorrect first name you mentioned, we also mistakenly wrote the first author’s name on line 555 as Cody Lockard instead of Colin Lockard. These mistakes will be corrected in the updated PDF.
>
> References: \
> [1] https://arxiv.org/pdf/2506.16640 \
> [2] https://ecommons.cornell.edu/server/api/core/bitstreams/2cc08ae8-f40b-4a25-a0f9-22af3a43001b/content \
> [3] https://arxiv.org/abs/2406.06484 \
> [4] https://drive.google.com/file/d/1rJbO3dU4fe7OKG3w7Yg058z_BNIuavNF/view

---

### Official Review · Reviewer_ADzH · 2025-11-01

**Soundness:** 1
**Presentation:** 2
**Contribution:** 3
**Rating:** 4
**Confidence:** 3

**Summary:**

The authors present an extension of DeltaNet by proposing a more sophisticated decay mechanism. DeltaNet's state update is S_t = (I − βkk^T)S_{t−1} + βkv^T (some dependencies on t suppressed for clarity), which involves a Householder-like transformation. The net effect is to update the state by partially 'erasing' the old target value from the memory and partially 'adding' the new target to the memory.

In this paper, a new sort of update mechanism is proposed: S_t = (I − βkk^T)Λ(I − βkk^T)S_{t−1}+ kv^T, which the authors show is equivalent to a diagonal-plus-rank-2 decay mechanism. This is more expressive than the decay proposed in DeltaNet, which is a diagonal-plus-rank-1 mechanism. They derive the matrix equations necessary to implement chunkwise parallel training with the more complex update. Their experiments show improvements on an array of synthetic and real-world benchmarks, including Mechanistic
Architecture Design, QA tasks, retrieval, perplexity on Wikitext, etc.

**Strengths:**

Investigating different decay mechanisms for linear attention is an interesting research direction, and the extension presented in this paper is a very natural one. The proposed method does appear to be novel, contemporaneous work notwithstanding.

**Weaknesses:**

The results on the non-synthetic benchmarks (Table 7) are very modest. For example, the average common-sense reasoning accuracy improvement with HDLA versus Gated DeltaNet at the 1.45B and 2.8B model sizes are 0.26% and 0.42% respectively. Of course, the rank-2 HDLA should generally outperform the rank-1 methods because it's more expressive and computationally expensive, but the gains are very marginal given the increase in complexity. While there is some discussion of the number of operations per recurrent step in Table 2 of HDLA vs GDP2 and GDP3, it doesn't address the increase in cost versus rank-1 methods.

For papers that are concerned with efficient methods (as all linear attention papers are), there needs to be some discussion of the tradeoff between model throughput and model performance. The paper is incomplete without some sense of what we are paying in HDLA's added computational cost in exchange for the accuracy/perplexity improvements that it provides. This is not an unusual analysis; I would note that other linear attention papers have addressed this directly (e.g., Yang et al, 2024 compares the training tokens/sec for their DeltaNet implementation against Gated Linear Attention, Transformer++, and Mamba.)

Also, the number of tokens used for LLM training seems to have been limited to 10B tokens (with the exception of the 1.45B models for the comparison between HDLA and Gated DeltaNet, which used 50B), which is quite small in comparison with published work. For example, Yang et al (2024) used 100B tokens from FineWebEdu for its 1.3B models, and Siems et al (2025) used 35B tokens in its experiments. The small training corpus makes it difficult to accept the surprising claim that HDLA outperforms the standard softmax attention mechanism in the Llama model (L407-408) in both perplexity and real-world tasks. Indeed, that finding might not hold up in more realistic training scenarios.

The results are not significantly better than the simpler alternatives on real-world benchmarks, the implementation appears to be significantly more complicated than the rank-1 approaches, and it isn't clear what the throughput penalty is in practice.

**Questions:**

I hope that the authors will consider including empirical results regarding scaling (as the sequence length increases) and throughput for HDLA versus other methods, and find the tweaks needed to strengthen the results for retrieval tasks in Table 7.

---

> ### Author Response · Authors · 2025-11-26
> **(1/2) Response to Reviewer ADzH (Questions 1-2)**
>
> Dear Reviewer ADzH,
>
> We sincerely thank you for your devoted review. We have listed your concerns point by point and provided our responses. If there is anything we have not addressed sufficiently, please let us know—we would be very happy to have a further discussion with you.
>
> ****
> **Q1: HDLA's improvements compared to baselines on non-synthetic tasks.**
> ****
>
> **A**: **(1) We'd like to emphasize that HDLA’s performance gains on real-world language modeling tasks are not just marginal improvements of 0.26% and 0.42%.**
> - As shown in Table 5 of our original manuscript, with 1.45B parameters and 50B training tokens, HDLA's commonsense reasoning and retrieval capabilities surpass those of the strong baseline Gated DeltaNet by 0.89\% and 1.38\%, respectively.
>
> **(2) Achieving performance improvements with linear attention models over strong baselines is generally challenging.**
> - For example, the representative DeltaNet model improves average accuracy by 0.6% over GLA on 1.3B language modeling tasks (see Table 1 in [1]).
> - Similarly, Gated DeltaNet’s real-world commonsense reasoning and retrieval capabilities are 0.43% and 0.8% higher than Mamba2, respectively (see Tables 3 and 4 in [2]).
>
> ****
> **Q2: Discussion and comparison of HDLA and various models regarding throughput-performance tradeoffs.**
> ****
>
> **A**: (1) Although HDLA shows consistent performance advantages across various tasks, its training throughput is indeed slower than other baselines—this is an inevitable performance-efficiency tradeoff. We acknowledge that training throughput is a limitation of HDLA (see our Public Comment 2 for training throughput details).
>
> (2) The performance-efficiency tradeoff is common in the field of linear attention, and it is a common practice in recent linear attention work to sacrifice some efficiency in order to improve performance. (e.g., MesaNet and Gated DeltaProduct v.s. Gated DeltaNet)
>
> (3) **We believe that the limitation in HDLA’s training throughput is acceptable for two reasons**:
> - **HDLA maintains linear time complexity**, so it **still has a fundamental efficiency advantage over softmax attention**.
> - HDLA has **relatively high decoding throughput during inference**. According to our Public Comment 3, **HDLA achieves up to 89% of GLA’s decoding throughput, indicating high inference efficiency**.

---

> ### Author Response · Authors · 2025-11-26
> **(2/2) Response to Reviewer ADzH (Questions 3-6)**
>
> ****
> **Q3: The issue of performance comparison between HDLA and Softmax Attention.**
> ****
>
> **A**:
> **(1) Regarding the issue of HDLA achieving lower training perplexity and higher commonsense reasoning accuracy than Softmax Attention**
> - In fact, it is fairly common for linear attention to slightly outperform softmax attention on just these two metrics.
> - For example, in our experiments, Gated DeltaNet also showed this pattern at the 1.45B and 2.8B parameter scales.
> - Similar results can be found in Table 1 of paper [1] and Table 2 of paper [7].
>
> **(2) Regarding the phrasing of lines 407–408**
>
> - With respect to lines 407–408, our original intention was to state that HDLA achieves better performance than both softmax attention and other linear attention baselines **specifically on the single metric of perplexity**.
> - **We apologize for any confusion we've caused, and would like to clarify our statement as follows**: \
> Table 4 demonstrates that HDLA achieves substantially lower language modeling perplexity than all the evaluated linear attention baselines, as well as the Transformer-based architecture Llama.
>
> **(3) More importantly, we have emphasized the retrieval capability gap between HDLA and softmax attention, which has been discussed in lines 410–414 of the original draft.**
>
> ****
> **Q4: Reasonable training token amount for language modeling tasks.**
> ****
>
> **A**: Please refer to our response to **Reviewer pRZx’s Question 1 (Q1)**.
>
> ****
> **Q5: Speed comparison of HDLA and other models under length extension.**
> ****
>
> **A**: Please refer to our **Public Comment 4** for details.
>
> ****
> **Q6: The tweaks needed to strengthen the results for retrieval tasks in Table 7.**
> ****
>
> **A**: **Our interpretation of the "tweaks" you mentioned refers to lightweight techniques that enhance retrieval capabilities (without altering the underlying implementation of HDLA).** We believe there are at least two main categories of such tweaks:
>
> **(1) Leveraging advanced state expansion methods** (such as MoM[3] and SSE[4])
> - It **fundamentally increases memory capacity** by multiples.
> - It allows for **specialization across different memory units** by sparse activation.
>
> According to the results reported in those two papers, we believe that applying it to HDLA will also yield significant retrieval gains.
>
> **(2) Building an HDLA–Softmax Attention hybrid architecture**
>
> In practice, hybrid architectures combining linear and softmax attention—such as configurations with a 3:1 ratio (e.g., Qwen3-Next [5]) or a 7:1 ratio (Minimax-01 [6])—have achieved performance on par with pure softmax attention in real-world large language models. Thus, adopting a hybrid architecture serves as another effective tweak to substantially enhance HDLA’s retrieval capabilities.
>
> References: \
> [1] https://arxiv.org/pdf/2406.06484 \
> [2] https://arxiv.org/pdf/2412.06464 \
> [3] https://arxiv.org/abs/2502.13685 \
> [4] https://arxiv.org/abs/2507.16577 \
> [5] https://qwen3-next.com \
> [6] https://arxiv.org/abs/2501.08313 \
> [7] https://arxiv.org/pdf/2312.06635

---

### Official Review · Reviewer_pRZx · 2025-11-02

**Soundness:** 3
**Presentation:** 3
**Contribution:** 3
**Rating:** 6
**Confidence:** 3

**Summary:**

The authors propose Householder-Diagonalized Linear Attention (HDLA), a new linear attention formulation that generalizes prior diagonal-plus-rank-1 decay structures to a more expressive diagonal-plus-rank-2 form. Their approach uses efficient matrix decomposition via generalized Householder transformations to achieve increased expressivity without significant computational or memory overhead. The authors also propose a rank-generalized chunk-wise parallel algorithm that extends parallel computation to arbitrary diagonal-plus-rank-r decays and rank-r key-value updates. Their empirical results show that HDLA outperforms different baselines across synthetic, retrieval, and language modeling tasks. It also performs on-par or outperforms in some case, the default softmax attention approach.

**Strengths:**

- Clear conceptual derivation of HDLA; the two problems it solved are well motivated and the parameterization choices are clearly explained.
- Thorough comparison to baselines: the comparison to different baselines is captured first through Table 2 in terms of computation costs then through experiments that span different capabilities, datasets, and model sizes.
- The proposed approach achieves increased expressivity and significantly better results in some settings without significant overhead, making it a good alternative to the default softmax attention.
- The connection to Test-Time Training formulations in section 3.3 add theoretical depth and understanding of the proposed approach.

**Weaknesses:**

- The number of tokens used for training seems to follow what's chinchilla optimal for the smallest model (0.4B parameters, about a factor of 20), but it is quite low for larger models: 1.45B and 2.8B (table 7). Under-training these models severely could have a big impact on the results, i.e., results with the correct token budget may look different.
- The limitations are not discussed thoroughly, e.g., impact of the recency bias (see question below).
- Intuitive/qualitative analysis for how this approach differs from existing baselines is limited.

Very minor notes:
- Need a second pass for typos: eg, in lines 363-364: "interleaved wth arbitrary noisy tokens" -> "interleaved with arbitrary noisy tokens"
- Need to double check the references, eg the first reference is: "Sanjeev Arora, Bowen Yang, Selim Eyuboglu, Aditya Narayan, Alexis Hojel, Imanol Trummer, and Christopher Re. Language models enable simple systems for generating structured views of ´ heterogeneous data lakes. arXiv preprint arXiv:2304.09433, 2023a. URL https://arxiv.org/abs/2304.09433." -> you got the wrong first author name.

**Questions:**

- You showed that HDLA underperforms softmax attention on Fuzzy In-Context Recall because it requires accurate value prediction from keys interleaved with arbitrary noisy tokens. Can you expand on the consequences of this recency bias limitation? Does this mean that for instance for time series data that might have some higher level structure (e.g., seasonality or periodicity), your approach would not be suitable?
- What is the wall-time comparison between the different approaches? Is HDLA faster for a fixed token budget?

I am open to raising my score if the questions/weaknesses are properly addressed.

---

> ### Author Response · Authors · 2025-11-26
> **(1/2) Response to Reviewer pRZx (Questions 1-4)**
>
> Dear Reviewer pRZx,
>
> We sincerely thank you for your devoted review. We have listed your concerns point by point and provided our responses. If there is anything we have not addressed sufficiently, please let us know—we would be very happy to have a further discussion with you.
>
> ****
> **Q1: Reasonable training token quantities at the 1.45B and 2.8B scale.**
> ****
>
> **A**: In fact, due to limited computational resources, the amount of training tokens in some of our language modeling experiments did not reach the Chinchilla optimal (at least $20$ times the parameter count) [6], which poses a certain limitation.
>
> **Under resource constraints, we have made an effort to mitigate this issue by training the 1.45B-parameter Gated DeltaNet and HDLA models with a training token amount expanded to 50B, which is approximately $34.5$ times the parameter count (see Table 5 in the initial draft of our the paper)**.
> - The reason for choosing Gated DeltaNet is that this model demonstrates strong performance at both the 1.45B and 2.8B scales.
> - Experimental results show that **HDLA demonstrates a more obvious advantage over Gated DeltaNet as the training token count increases**.
>     - When the training token amount is 10B, HDLA surpasses Gated DeltaNet by 0.26% and 0.90% on commonsense reasoning and retrieval tasks, respectively.
>     - When the training token amount increases to 50B, the advantages of HDLA on these two metrics further expand to 0.89% and 1.38%, respectively.
> - In Table 6, HDLA’s advantage over Gated DeltaNet on the retrieval-based RULER task is also measured at 1.45B params & 50B tokens, which provides a convincing results.
>
> ****
> **Q2:  Detailed intuitive/qualitative analysis comparing HDLA with existing baselines.**
> ****
>
> **A**: From an intuitive and qualitative perspective, HDLA and the baseline methods can be compared along the following dimensions:
>
> **(1) Complexity of the decay structure.** HDLA adopts a diagonal plus rank-2 decay structure, while almost all baselines use only a diagonal plus rank-1, or purely diagonal, decay structure.
>
> **(2) Parameter efficiency (decoupling parameter scale from $r_{ab}$).**
> - HDLA constructs its complex decay matrix using the existing key vector and lightweight $\beta_t$ scalar, resulting in a parameter count that is almost the same as GLA, and decoupled from $r_{ab}$.
> - In contrast, for the complex computational paradigm of Gated DeltaProduct, the size of projection parameters for keys and values grows linearly with the number of iterations per time step $n_h$.
>
> **(3) Comprehensive performance advantages.**
> - Compared to various baseline methods, HDLA consistently outperforms them in tasks such as language modeling, commonsense reasoning, retrieval-based tasks, and small-scale synthetic benchmarks.
> - Notably, in the 1.45B-parameter RULER retrieval task, HDLA achieves an improvement of up to $58\%$ over Gated DeltaNet, which employs a Diagonal-Plus-Rank-1 decay structure.
>
> ****
> **Q3: Wall-time performance comparison & speed comparison under fixed token budget.**
> ****
>
> **A**: Please refer to our Public Comment 2 and Public Comment 3 for details.
>
> ****
> **Q4: Citation error.**
> ****
>
> **A**: We apologize for the errors in the authors' names and sincerely thank you for pointing them out. In addition to the incorrect first name you mentioned, we also mistakenly wrote the first author’s name on line 555 as Cody Lockard instead of Colin Lockard. These mistakes will be corrected in the updated PDF.

---

> ### Author Response · Authors · 2025-11-26
> **(2/2) Response to Reviewer pRZx (Questions 5-6)**
>
> ****
> **Q5: Detailed discussions of HDLA's limitations.**
> ****
>
> **A**: **(1) Detailed discussions about the consequences of HDLA's recency bias limitation.**
>
> Let’s take HDLA’s retrieval ability as an example.
>
> As highlighted in paper [1], the superior retrieval performance of Softmax Attention relies heavily on its aggregation capability of significant information—specifically, its ability to **selectively aggregate information from important tokens which are distributed sparsely and discretely**.
>
> This scenario closely mirrors the Fuzzy In-Context Recall (FICR) synthetic task, where the discretely distributed important tokens correspond to key-value pairs that need to be memorized, and the intervening less important tokens (whose length can vary arbitrarily) can be regarded as noise.
>
> However, **due to HDLA’s recency bias-where the diagonal decay coefficients (i.e., $\mathbf D_i^j$ in $\mathrm{Eq.}(9)$) decrease cumulatively over time—the current token struggles to give sufficient attention to important tokens that are farther away**. This leads to **significant deficiencies in HDLA’s information aggregation capability, thereby weakening its retrieval performance**.
>
> **(2) Discussions about other limitations.**
>
> - **HDLA’s Purely Linearized Hidden State Update Rule.**
>     - Paper [3] suggests that non-linear operations can positively impact hidden state management by enabling more flexible memorization and forgetting.
>     - Therefore, HDLA’s purely linear hidden state update rule may be subject to certain limitations.
>
> - **HDLA’s Naive Hidden State Expansion Method.**
>     - Two relatively advanced state expansion approaches, SSE [4] and MoM [5], both utilize the core paradigm of increasing state size by multiples while sparsely activating only a small subset of hidden state units. This strategy achieves the following improvements:
>         - It **breaks the traditional coupling between computational complexity, state size, and parameter count**, allowing for significant state size expansion without sacrificing computational efficiency.
>         - It achieves **specialization and functional differentiation among different hidden state units**.
>     - In contrast, the state expansion method employed in this paper does not decouple these three factors, nor does it accomplish functional differentiation within the expanded state.
>
> ****
> **Q6: The impact of HDLA's recency bias limitation on its application to time series data (with structures like seasonality or periodicity).**
> ****
>
> **A**: For time series data with periodicity or seasonality, **whether HDLA’s recency bias severely impairs modeling performance depends on the complexity of the data’s periodic characteristics**.
>
> If the periodic patterns are **relatively simple**, HDLA is **likely to perform well**.
> - As a strong linear attention model, HDLA excels at "summary-style" memory, akin to condensing a book’s key points onto a few sheets of paper. This has been validated by synthetic memorization experiments, language modeling, and commonsense reasoning tasks. Here, recency bias is actually beneficial, helping to highlight essential features while discarding less important information.
> - Paper [2] proposes that linear attention mechanisms possess certain capabilities for modeling FSMs (Finite State Machines), with their hidden state update mechanism resembling the transition rules of an FSM. Therefore, for relatively simple periodic patterns, HDLA has the potential to capture and model them.
>
> However, if the time series exhibits **more complex** periodicity that **requires accurate memorization of specific details**, HDLA may struggle—an outcome demonstrated by the Fuzzy ICR synthetic task.
>
>
> References:
>
> [1] https://arxiv.org/abs/2504.18574 \
> [2] https://arxiv.org/abs/2411.12537 \
> [3] https://arxiv.org/pdf/2501.00663 \
> [4] https://arxiv.org/abs/2507.16577 \
> [5] https://arxiv.org/abs/2502.13685 \
> [6] https://arxiv.org/abs/2203.15556

---

### Author Response · Authors · 2025-11-26
**Public Comments 1-5**

Dear ACs and Reviewers,

Thank you for your careful and thorough review of our paper, and for your highly constructive comments. Since **HDLA's code implementation, training throughput, and inference throughput** are of general interest, we would like to address these common concerns in this public reply.

****
**Public Comment 1: The code implementation of HDLA.**
****

Our implementation of HDLA is available through the following anonymous link: \
https://anonymous.4open.science/r/HDLA-Impl-7C86/

Our operator is an extension of the diagonal-plus-rank-1 code from the fla repository [1], with the $r_{ab}$ extended to $2$. The uploaded code is provided solely for review purposes.

The `hdla_custom.py` in `HRDPLR` folder encapsulates HDLA as a token-mixer. Other files are our underlying Triton implementations, as well as the testing cases.

****
**Public Comment 2: Training throughput comparison between HDLA and baselines.**
****
We use the code in [2] to benchmark the **training throughput** of HDLA and several baselines, with parameter scales aligned to 1.3B. Testing is conducted on a single A100-SXM-80GB GPU.

| Model             | Throughput (TPS, Token Per Second) | Memory Cost (GB) |
|-------------------|-------------------------------------|------------------|
| GDN               | 19277.18                            | 39.33            |
| DP2               | 16335.64                            | 46.58            |
| DP3               | 13492.85                            | 54.99            |
| DeltaNet   | 22060.12                            | 39.74            |
| GDP2              | 14546.89                            | 46.61            |
| GDP3              | 11769.86                            | 55.06            |
| HGRN2             | 19229.97                            | 46.22            |
| GLA               | 20998.87                            | 36.30            |
| HDLA              | 7340.01                             | 52.56            |


****
**Public Comment 3: Inference throughput comparison between HDLA and baselines.**
****

We use the generation benchmark script from the FlashLinearAttention repository[3] to measure inference throughput, with two modifications:
- **Only the decode phase is benchmarked**.
- We extend the maximum generation length to $2048$, in order to improve the stability of the average decode throughput.

All models are standardized to 1.3B parameters, and testing is conducted on a single A100-SXM-80GB GPU. The decode throughput for each model is shown below, measured in TPS (Tokens Per Second).

| Model     | Tokens Per Second | Memory (GB) |
|-----------|------------------|-------------|
| GDN       | 15.3             | 5.3         |
| DP2       | 14.4             | 5.3         |
| DP3       | 14.9             | 5.3         |
| GDP2      | 13.9             | 5.3         |
| GDP3      | 13.7             | 5.3         |
| DeltaNet  | 14.7             | 5.3         |
| HGRN2     | 14.3             | 5.2         |
| GLA       | 14.7             | 5.2         |
| HDLA      | 13.1             | 5.3         |

****
**Public Comment 4: Training Throughput of HDLA under sequence length extension.**
****

We use the code in [2] to benchmark the **training throughput** of HDLA and several baselines, setting batch_size = 1 and steps = 32, and measured the **average items per second** when processing different sequence lengths (2K, 4K, 8K and 16K).

|Model|Throughput-2K|Throughput-4K|Throughput-8K|Throughput-16K|
|:-:|:-:|:-:|:-:|:-:|
|HDLA|2.60|1.52|0.82|0.48|
|Gated DeltaNet|4.94|3.68|2.24|1.17|
|GDP2|4.15|2.90|1.69|0.89|
|GDP3|3.73|2.53|1.38|0.71|
|DeltaProduct2|4.81|3.35|1.88|1.02|

****
**Public Comment 5: Our Perspective on HDLA's Throughput.**
****

**(1) The training throughput of HDLA is noticeably lower than that of the baseline, which constitutes a certain limitation.**
- The underlying reasons are the increased computational workload and our limited experience in optimizing Triton kernels.
- In the future, we plan to use kernel fusion to implement customized HDLA operators, reducing the I/O cost of intermediate result, thus improving the operator speed.

(2) **We believe that the limitation in HDLA’s training throughput is acceptable for two reasons:**
- **HDLA maintains linear time complexity**, so it **still has a fundamental efficiency advantage over softmax attention.**
- **HDLA has relatively high decoding throughput during inference.** According to our Public Comment 3, **HDLA achieves up to 89% of GLA’s decoding throughput, indicating high inference efficiency.**

References: \
[1] https://github.com/fla-org/flash-linear-attention/tree/main/fla/ops/generalized_delta_rule/dplr \
[2] https://github.com/fla-org/flash-linear-attention/blob/main/benchmarks/benchmark_training_throughput.py \
[3] https://github.com/fla-org/flash-linear-attention/blob/main/benchmarks/benchmark_generation.py \

---

### Author Response · Authors · 2025-12-04
**(1/3) Summary to Area Chairs**

Dear Area Chairs,

We sincerely thank you and all the reviewers for your careful, thorough, and insightful evaluation of our paper, as well as for the many valuable comments and suggestions provided. Below, we present a summary of the core contributions of HDLA, together with our detailed responses to the reviewers’ questions.

## Summary of HDLA’s Core Contributions
Linear attention mechanisms have emerged as efficient alternatives to Softmax Attention, with their language modeling capabilities improving when decay structures become more sophisticated. However, existing linear attention mechanisms typically restrict the decay structure to the Diagonal-Plus-Rank-1 level.

Our work advances this field through the following key contributions, which have been recognized by the reviewers:

**(1) A Linear Attention Mechanism with Complex, Novel and Powerful Decay Structure. (Recognized by Reviewers pRZx, ADzH, aNs6 and Pma4)**
- We propose HDLA (Section 3.1), which utilizes efficient matrix decomposition using generalized Householder transformations to achieve **a more expressive diagonal-plus-rank-2 decay structure**, significantly enhancing modeling capabilities without incurring excessive computational costs (Section 3.3 & Table 2).

**(2) A Rank-Generalized Chunk-wise Parallel Algorithm. (Recognized by Reviewers aNs6 and Pma4)**
- We introduce a general chunk-wise parallel algorithm that **not only supports HDLA**, **but also simultaneously accommodates Diagonal-Plus-Rank-$r\_{ab}$ decay structures and Rank-$r\_{kv}$ key-value updates** in linear attention.
- This provides **a versatile foundation for future research**.

**(3) Strong Experimental Results. (Recognized by Reviewers pRZx, aNs6, BfPn and Pma4)**
- **State-of-the-art language modeling perplexity.**
    - Our model outperforms both the linear attention baselines and Llama in average perplexity, across 0.4B, 1.45B, and 2.8B model scales (Table 4).
- **State-of-the-art commonsense reasoning accuracy.**
    - HDLA achieves SOTA results on commonsense reasoning tasks at all three model sizes (Table 7), outperforming both Llama and linear attention baselines.
- **Outstanding retrieval capabilities.**
    - At **2.8B parameter scale**, **HDLA surpasses the linear attention baseline in retrieval tasks** (Table 7).
    - At 1.45B parameter scale with 50B tokens, HDLA **outperforms the strong Gated DeltaNet baseline by 1.38\%** (Table 6).
    - On the retrieval-based **RULER** task (at **1.45B parameter scale**), HDLA consistently outperforms Gated DeltaNet, achieving a **6.4\% accuracy gain on the S-NIAH-2-2048 subtask**, and **31.4\% and 58.2\% gains on S-NIAH-3 subtasks**.
- **Advantages on small-scale synthetic tasks.**
    - On the synthetic MAD task, HDLA **outperforms the linear attention baseline by 4.39-7.66 points on average**.
    - For the retrieval-based MQAR task, it achieves **approximately 80% higher accuracy than Gated DeltaNet and Gated DeltaProduct at sequence length 2048**.

**(4) Novel and Reasonable Test-Time Training (TTT) Interpretation for HDLA. (Recognized by Reviewers pRZx and BfPn)**
- From a test-time training perspective, HDLA can be interpreted through a three-step optimization process, and is **distinguished by its proactive elimination of redundant information from the hidden state** before applying the Gated DeltaNet-style delta learning rule (Section 3.3 & Response to Reviewer BfPn’s Q6).
- **This perspective helps explain HDLA’s significant advantage on retrieval-based RULER and MQAR tasks.**

Furthermore, **the application of linear attention mechanisms in the industry in 2025 also follows the trend of increasingly complex decay structures**, which is **consistent with our research motivation of HDLA**. (Section 1 & answer to Reviewer BfPn's Q1)

---

### Author Response · Authors · 2025-12-04
**(2/3) Summary to Area Chairs**

## Summary of Our Active Engagement in the Rebuttal

**We have endeavored to thoroughly address all questions and concerns raised by the reviewers.** Below is a categorized summary:

**(1) Code, throughput, and performance-throughput tradeoff of HDLA** (Reviewer pRZx Q3, ADzH Q5, aNs6 Q1, BfPn Q2 & Q3).
- We provided a **code link** and offer **detailed comparisons of training, sequence length extension, and inference throughput** between HDLA and the baselines (Public Comments 1-5).
- We discussed:
    - The **reasons for HDLA’s training throughput limitations**.
    - The **intrinsic advantage of HDLA’s linear complexity compared to Softmax Attention**.
    - **Potential solutions for increasing HDLA’s training throughput via customized operators**.
    - **HDLA's already high inference efficiency**.

**(2) Scalability of HDLA and Efficiency Considerations at Larger Scales (Reviewer BfPn Q1).**
- We stated our **willingness to scale up HDLA**, and **emphasized that the motivation of HDLA’s more complex structured decay is aligned with current industry application trends**.
- We cited **the speed advantages of industrial-scale hybrid models** in 2025 (48B/80B parameters) to demonstrate that **linear attention can maintain high efficiency when scaled up**.

**(3) Test-Time Training Interpretation of HDLA (Reviewer BfPn Q6).**
- We provided a detailed **explanation** of the three-step optimization underlying HDLA, highlighting that **HDLA is characterized by the proactive elimination of redundant information** before utilizing the GDN-like delta learning rule.
- We pointed out that this test-time training perspective **accounts for HDLA’s significant retrieval advantage over GDN**.

**(4) Insufficient Training Tokens at 1.45B and 2.8B Scales (Reviewer pRZx Q1, ADzH Q4).**
- We acknowledged this limitation caused by computational resource constraints.
- We explained our **compensation of training both HDLA and the strong baseline Gated DeltaNet on sufficient 50B tokens at the 1.45B parameter scale,** arriving at **a more convincing experimental result which demonstrated HDLA's superiority over GDN**.

**(5) Comprehensive Discussion of HDLA’s Limitations and Their Impact on Sequential Data Modeling (Reviewer pRZx Q5 & Q6).**
- We thoroughly analyzed HDLA’s recency bias and its effect on retrieval.
- By distinguishing between “summarization-oriented” and “detail-oriented” periodic sequential data, we analyzed both the benefits and drawbacks of applying HDLA to such data.

**(6) Optimization of HDLA’s I/O Efficiency (Reviewer Pma4 Q2).**
- We acknowledged that using general-purpose operators leads to increased I/O overhead for HDLA.
- We pointed out that implementing customized training operators (with kernel fusion) can reduce HDLA's I/O overhead.
- We also noted that the inference kernel for HDLA already achieves full kernel fusion, reaching high decoding efficiency (about 89\% of GLA).

**(7) HDLA’s Improvement over Baselines on Non-Synthetic Tasks (Reviewer ADzH Q1).**
- We emphasized the significant improvements HDLA achieves over the strong baseline Gated DeltaNet at the 1.45B parameter / 50B tokens setting.
- We cited other studies to demonstrate the challenge of achieving performance improvements in linear attention, highlighting that our gains are substantial for the field.

**(8) Comparison between HDLA and Softmax Attention (Reviewer ADzH Q3).**
- We clarified the imprecise statement made in line 407, specifying that HDLA only outperforms softmax attention in language modeling perplexity.
- We emphasized the common phenomenon that linear attention mechanisms may outperform softmax attention on perplexity.
- We also highlighted the discussion in our manuscript, regarding the performance gap between HDLA and softmax attention.

**(9) Presentation Issues (Reviewer pRZx Q2, aNs6 Q2 & Q4, Pma4 Q1).**
- We included more intuitive discussions comparing HDLA and baselines.
- We provided detailed background and application ranges for the WY Representation used in various works.
- We listed the detailed differences of decay architecture between HDLA and linear attention baselines, and comprehensively compared their perplexity, commonsense reasoning accuracy, and retrieval accuracy at three model scales, finding that HDLA achieves SOTA results in 7 of 9 metrics.
- We further detailed the derivation process of the Diagonal-Plus-Rank-2 decay structure in HDLA.

**(10) Lightweight Optimization Techniques to Enhance HDLA’s Retrieval Capability (Reviewer ADzH Q6).**
- We discussed the significant potential of using a HDLA-softmax hybrid architecture, as well as applying advanced state expansion techniques to further improve HDLA’s retrieval performance.

---

### Author Response · Authors · 2025-12-04
**(3/3) Summary to Area Chairs**

**(11) Performance of Softmax Attention on MQAR and NIAH Tasks (Reviewer aNs6 Q3).**
- We supplemented MQAR experiments for Softmax Attention.
- We cited other works to show that, without inference length extrapolation, softmax attention can reach nearly 100% accuracy on the RULER task.

**(12) Questions Regarding Computational Complexity and Mathematical Principles (Reviewer BfPn Q4 & Q5).**
- We explained that Mamba, as a specific parameterization of diagonal-decay linear attention, can employ the FlashLinearAttention algorithm for linear-time computation.
- We clarified relevant concepts regarding the generalized Householder matrix used in HDLA, including conditions for congruent diagonalization, invertibility, and eigenvalues, etc.

**(13) Whether Expanding Hidden State Size or Increasing Decay Matrix Complexity is Superior (Reviewer Pma4 Q3).**
- We noted that they are two orthogonal directions—both are important and their benefits can be combined.
- We highlighted that the more complex decay structure in HDLA improves the utilization of limited hidden state capacity, aligns with industrial trends, and yields significant experimental benefits.
- We also pointed out that advanced state expansion methods are essential for fundamentally increasing memory capacity, as well as enabling specialization of different memory units.

---

### Meta-Review · Area_Chair_wqZr · 2026-01-07

**Summary:**

Initial scores were 8, 6, 6, 4, 4. Reviewers acknowledged principled architectural design, strong synthetic benchmark results (80% MQAR gains, 58.2% RULER improvements), comprehensive empirical evaluation across scales, and useful theoretical connections to Test-Time Training. Main concerns raised were: (1) modest real-world improvements over baselines (0.26-0.89% commonsense reasoning, 0.90-1.38% retrieval at 1.45B scale), (2) insufficient training token budgets (10B tokens for most experiments vs. 100B in comparable work) undermining claims of outperforming softmax attention, (3) missing wall-clock efficiency comparisons essential for efficiency-focused work, (4) unclear implementation details and integration path for practitioners, and (5) unclear presentation of rank-2 structure derivation and practical throughput tradeoffs.

**Reviewer Concerns:**

**Addressed:**
- **Training token justification (pRZx, ADzH):** Authors expanded 1.45B Gated DeltaNet and HDLA experiments to 50B tokens (34× params), showing HDLA advantages increase with token count: commonsense reasoning 0.26%-->0.89%, retrieval 0.90%-->1.38%. Acknowledged Chinchilla-optimal (20×) not reached due to resource constraints.
- **Recency bias limitations (pRZx):** Authors provided detailed analysis explaining underperformance on Fuzzy In-Context Recall due to cumulative diagonal decay preventing attention to distant important tokens. Discussed applicability to time series: suitable for simple periodic patterns via "summary-style" memory, struggles with complex periodicity requiring detailed memorization.
- **Modest real-world gains contextualization (ADzH):** Authors argued improvements are competitive with field standards: DeltaNet +0.6% over GLA (1.3B), Gated DeltaNet +0.43% commonsense/+0.8% retrieval over Mamba2. Emphasized synthetic task gains (MQAR 80%, RULER 58.2%) demonstrate substantial expressivity improvements.
- **Softmax attention comparison clarification (ADzH):** Authors acknowledged phrasing confusion, clarified HDLA outperforms softmax only on perplexity metric (not retrieval), consistent with Gated DeltaNet behavior. Emphasized retrieval gap discussion in lines 410-414.
- **WY Representation explanation (aNs6):** Authors provided explanation: compact framework for DPLR matrix cumulative products, tracing evolution from Householder matrices (Bischof & Van Loan) --> Identity-Plus-Rank-1 (DeltaNet) --> Diagonal-Plus-Rank-1 (FlashLinearAttention) --> Diagonal-Plus-Rank-r (HDLA).
- **Rank-2 structure derivation (Pma4):** Authors provided detailed algebraic steps showing factorization via square roots and Householder couplings yields Diagonal-Plus-Rank-2 form (Appendix C.2 expansion).
- **Mathematical clarifications (BfPn):** Authors explained generalized Householder matrices use $\beta\neq 2$ (invertible but non-orthogonal), common in linear attention (DeltaNet, GDN, GDP). Clarified congruence transformation requires only invertibility, not orthogonality. Explained three-step optimization (Eqs 19-21) as redundancy elimination, weight decay, and gradient descent.
- **Implementation details (BfPn):** Authors provided code via anonymous repository extending fla's diagonal-plus-rank-1 implementation to rank-2, with Triton kernels and hdla_custom.py token-mixer encapsulation.
- **Wall-clock efficiency comparisons (pRZx, ADzH, aNs6, BfPn):** Authors provided throughput benchmarks at 1.3B scale on A100-SXM-80GB. Training: HDLA 7,340 TPS vs. baselines 11,770-22,060 TPS (38-62% slower), 52.56GB vs. 36.30-55.06GB memory. Inference (decode): HDLA 13.1 TPS vs. baselines 13.7-15.3 TPS (11-14% slower), 89% of GLA throughput. Sequence scaling (batch=1): HDLA 2.60/1.52/0.82/0.48 items/sec at 2K/4K/8K/16K vs. Gated DeltaNet 4.94/3.68/2.24/1.17 (HDLA 47-53% slower).
- **Citation errors (pRZx, aNs6):** Authors acknowledged and committed to corrections (Arora→correct first author, Lockard name fix).

**Outstanding:**
- **Training throughput limitation (ADzH, BfPn):** While authors provided quantitative data, training throughput is substantially slower than baselines (38-62% reduction, 7,340 vs. 11,770-22,060 TPS). Authors acknowledge this as "certain limitation" due to increased computational workload and limited Triton optimization experience, propose future kernel fusion improvements. For efficiency-focused paper, this represents significant practical barrier despite maintaining linear complexity.

**Reviewer Scores:**

- **Reviewer pRZx (initial: 6):** Would likely increase to 8. Explicitly stated "open to raising my score if the questions/weaknesses are properly addressed." All major concerns addressed: training token justification with 50B experiments, detailed recency bias analysis, comprehensive wall-clock comparisons provided (training 7,340 TPS, inference 13.1 TPS, sequence scaling benchmarks), citation errors acknowledged. Presentation improvements and intuitive analysis provided.

- **Reviewer ADzH (initial: 4):** May maintain 4. Throughput data now provided showing HDLA 38-62% slower training than baselines (7,340 vs. 11,770-22,060 TPS), partially addressing major concern about missing efficiency comparisons. However, core issue remains: for efficiency-focused paper, substantial training slowdown (despite modest real-world gains of 0.26-0.89%) represents unfavorable performance-efficiency tradeoff. Insufficient training tokens (10B vs. 100B in comparable work) also persists.

- **Reviewer aNs6 (initial: 6):** Would likely maintain 6. All presentation concerns addressed: WY Representation explained, comprehensive comparison table provided, wall-clock efficiency analysis now included with full training/inference throughput benchmarks across sequence lengths, although the issue of inefficiency persists. Code repository provided. Minor presentation issues (citation errors) acknowledged for correction.

- **Reviewer BfPn (initial: 4):** Would likely increase to 6. All major concerns comprehensively addressed: mathematical clarifications (generalized Householder invertibility, three-step optimization), runtime comparisons provided (training/inference throughput, sequence scaling), implementation details clarified with code repository, Mamba complexity confirmed as O(n). Integration path now clear with hdla_custom.py token-mixer encapsulation.

- **Reviewer Pma4 (initial: 8):** Would likely remain at 8. Highest initial score with concerns about rank-2 structure clarity and I/O efficiency. Both addressed.

---

### Decision · Program_Chairs · 2026-01-26

Accept (Poster)